# One dog's waste is another dog's wealth: A pilot study of fecal microbiota transplantation in dogs with acute hemorrhagic diarrhea syndrome

Arnon Gal[1]*, Patrick C. Barko[1], Patrick J. Biggs[2,3], Kristene R. Gedye[2], Anne C. Midwinter[2], David A. Williams[1], Richard K. Burchell[4], Paolo Pazzi[5]

1 Department of Veterinary Clinical Medicine, College of Veterinary Medicine, University of Illinois at Urbana-Champaign, Urbana, Illinois, United States of America, 2 Molecular Epidemiology & Public Health Laboratory, Infectious Disease Research Centre, School of Veterinary Science, Massey University, Palmerston North, New Zealand, 3 Bioinformatics and Statistics Group, School of Fundamental Sciences, Massey University, Palmerston North, New Zealand, 4 North Coast Veterinary and Referral Centre, Sunshine Coast, Queensland, Australia, 5 Department of Companion Animal Clinical Studies, Faculty of Veterinary Science, University of Pretoria, Pretoria, South Africa

☯ These authors contributed equally to this work.
* agal2@illinois.edu

**Data Availability Statement:** All 16S-rRNA Amplicon Sequencing files are available from the BioProject repository database (https://www.ncbi.

## Abstract

Canine acute hemorrhagic diarrhea syndrome (AHDS) has been associated in some studies with *Clostridioides perfringens* overgrowth and toxin-mediated necrosis of the intestinal mucosa. We aimed to determine the effect of a single fecal microbiota transplantation (FMT) on clinical scores and fecal microbiomes of 1 and 7 dogs with AHDS from New Zealand and South Africa. We hypothesized that FMT would improve AHDS clinical scores and increase microbiota alpha-diversity and short-chain fatty acid (SCFA)-producing microbial communities' abundances in dogs with AHDS after FMT. We sequenced the V3-V4 region of the 16S-rRNA gene in the feces of AHDS FMT-recipients and sham-treated control dogs, and their healthy donors at admission, discharge, and 30 days post-discharge. There were no significant differences in median AHDS clinical scores between FMT-recipients and sham-treated controls at admission or discharge (P = 0.22, P = 0.41). At admission, the Shannon diversity index (SDI) was lower in AHDS dogs than healthy donors (P = 0.002). The SDI did not change from admission to 30 days in sham-treated dogs yet increased in FMT-recipients from admission to discharge (P = 0.04) to levels not different than donors (P = 0.33) but significantly higher than sham-treated controls (P = 0.002). At 30 days, the SDI did not differ between FMT recipients, sham-treated controls, and donors (P = 0.88). Principal coordinate analysis of the Bray-Curtis index separated post-FMT and donor dogs from pre-FMT and sham-treated dogs (P = 0.009) because of increased SCFA-producing genera's abundances after FMT. A single co-abundance subnetwork contained many of the same OTUs found to be differentially abundant in FMT-recipients, and the abundance of this module was increased in FMT-recipients at discharge and 30 days, compared to sham-treated controls. We conclude in this small pilot study FMT did not have any clinical benefit.

nlm.nih.gov/bioproject/) under the accession number PRJNA629659 All bioinformatic analyses scripts are available from the Github database (https://github.com/pcbarko/Gal-Barko-K9-AHDS-FMT).

**Funding:** We declare that the funder (i.e., Massey University) provided support in the form of salaries for authors [AG, RB, PB, KG and AM], but did not have any additional role in the study design, data collection and analysis, decision to publish, or preparation of the manuscript. The specific roles of these authors are articulated in the 'author contributions' section.

**Competing interests:** Dr. Burchell's current affiliation did not alter our adherence to PLOS ONE policies on sharing data and materials. None of the authors have or had any conflict of interest with regards to conception, funding or execution of this study, analysis and interpretation of the results, or writing the manuscript of this study.

A single FMT procedure has the potential to increase bacterial communities of SCFA-producing genera important for intestinal health up to 30 days post-FMT.

## Introduction

Acute hemorrhagic diarrhea syndrome (AHDS) is a clinical syndrome characterized by a set of typical clinical signs (vomiting and watery bloody diarrhea) and clinicopathologic abnormalities (hemoconcentration with discordant serum protein levels) which are correlated with each other and associated with a particular clinical presentation (severe, peracute disease). Several bacterial etiologies, bacterial toxins, and dietary components have been associated with AHDS [1–11]. Evidence suggests that dogs with AHDS might also have gut dysbiosis [4, 10, 12], however, evidence differs with respect to the composition of gut bacterial communities, thus highlighting the inherent heterogeneity of this syndrome. To date, a direct cause-and-effect relationship between AHDS and specific bacterium, a state of gut dysbiosis, or bacterial toxins has not been established. Still, the histopathological pattern of mucosal injury in dogs with AHDS could support a central role for toxin-mediated necrosis of the intestinal mucosa [5, 8, 13]. If a state of dysbiosis precedes the onset of disease it might be expected that normobiosis would temporally ensue with the resolution of clinical signs. Yet, there is a gap in knowledge with regards to state of gut microbiota diversity in dogs with AHDS as well as after spontaneous clinical recovery.

Fecal microbial transplantation (FMT) is a medical procedure during which the engraftment of a healthy donor's gut microbiota is aimed at restoring the recipient's disrupted native microbiota, leading to stabilization of the microbiome and increased microbiota diversity. Fecal microbial transplantation has been best described in the treatment of people with recurrent *Clostridioides difficile* infections (RCI) in which the *C. difficile* overgrowth is associated with substantially decreased fecal microbial diversity and toxin-mediated intestinal mucosal necrosis [14–17] and in which FMT has astonishing high cure rates of 92%-98% [18–20]. There is limited literature about FMT in dogs. Fecal microbial transplantation has shown some promise by improving the diversity and composition of gut microbial communities and clinical signs of gastrointestinal dysfunction in dogs with both acute and chronic enteropathies [21–28]. However, there is a gap in knowledge with regards to the state of gut microbiota diversity in dogs in the short, intermediate and long term following FMT in general and in AHDS in particular.

The purpose of this study was to determine the effect of FMT on hospitalization time, AHDS score and the composition of fecal microbiota in dogs with AHDS. We hypothesized that administration of a single intervention of FMT at the proximal colon would result in earlier discharge from the hospital and improved clinical AHDS score, increase microbiota alpha-diversity and short-chain fatty acid (SCFA)-producing microbial communities' abundances in dogs with AHDS after FMT. Our specific aims were 1) to compare AHDS clinical scores and duration of hospitalization between groups, 2) to determine the temporal changes in the alpha diversity of recipient dogs with AHDS, before and after FMT, contrasted with those of their healthy donor dogs and sham-treated control dogs with AHDS; 3) to determine the temporal changes in beta diversity of the bacterial communities in the healthy donors, FMT recipient dogs with AHDS, and sham-treated control dogs with AHDS; and 4) to characterize the specific bacterial communities that are significantly different between the donor-recipient dogs and the sham-treated control dogs following FMT.

## Material and methods

### Animals

Seven and 1 client-owned dogs with clinical signs consistent with AHDS that met the inclusion and exclusion criteria for the study (see below) were prospectively enrolled between June 2016 to September 2017 from the Onderstepoort Veterinary Academic Hospital (South Africa) and Massey University Teaching Hospital (New Zealand), respectively (Table 1). Three and 1 clinically healthy donor dogs owned by hospital staff members from the Onderstepoort Veterinary Academic Hospital and Massey University Teaching Hospital, respectively, were enrolled as fecal donors (Table 1). The study was approved by the Massey University Animal Ethics Committee (MUAEC#15/74) and the University of Pretoria Animal Ethics Committee (#V100-16, S4285-15).

### Patient inclusion/exclusion criteria

Patients were included if they presented with acute onset of a large volume of watery bloody diarrhea of less than 3 days duration. Hemoconcentration (packed-cell volume or hematocrit

**Table 1. Demographic information of donors and AHDS dogs.**

| | Group** | Breed | Age (years) | Sex | Admission body Weight (kg) | Duration of diarrhea before presentation (days) | Diet | Duration of hospitalization (days) | AHDS index at admission | AHDS index on day 2* |
|---|---|---|---|---|---|---|---|---|---|---|
| **Donors** | | | | | | | | | | |
| Massey dog | (M1) | Mixed breed | 8 | FS | 28 | N/A | Proplan sport biscuits | | | |
| Onderstepoort dog 1 | (OP1) | Mixed breed | 9 | FS | 15.2 | N/A | Hill's™ Metabolic | | | |
| Onderstepoort dog 2 | (OP2) | Mixed breed | 8 | MN | 22.1 | N/A | Hill's™ Adult | | | |
| Onderstepoort dog 3 | (OP3) | Great Dane | 6 | MN | 54.2 | N/A | Montego Adult | | | |
| **Recipients** | | | | | | | | | | |
| Massey AHDS 1 | FT (M1) | French Bulldog | 3 | MN | 13 | 2 | Hill's™ z/d | 3 | 8 | 4 |
| Onderstepoort AHDS 1 | S | Dachshund | 5 | MN | 5.6 | 2 | Unknown | 3 | 4 | 0 |
| Onderstepoort AHDS 2 | S | Scottish terrier | 6 | MN | 10.8 | 1 | Unknown | 3 | 12 | 2 |
| Onderstepoort AHDS 3 | S | Mixed breed | 6 | FI | 4.0 | 0.5 | Bob Martin Pellets | 3 | 4 | 3 |
| Onderstepoort AHDS 4 | FT (OP1) | Yorkshire terrier | 12 | MN | 3.2 | 0.5 | Boiled chicken | 3 | 4 | 4 |
| Onderstepoort AHDS 5 | FT (OP2) | Dachshund | 8.5 | MN | 4.2 | 2 | Unknown | 3 | 9 | 2 |
| Onderstepoort AHDS 6 | S | Pekingese | 3 | FS | 4.6 | 0.5 | Unknown | 3 | 7 | 5 |
| Onderstepoort AHDS 7 | FT (OP3) | Dachshund | 5 | FS | 7.8 | 0.25 | Royal Canin^R pellets & chicken livers | 3 | 7 | 5 |

*Day 2 was the day of fecal transplant.

**FT: Fecal microbiota transplantation; S: Sham-treated control; (): Donor-FMT recipient pair.

MN: Male neutered; FI: Female intact; FS: Female sterilized.

at or above the high end of the reference interval) and either a low or low-normal serum total protein concentration were additional inclusion criteria. Patients enrolled in the study were examined by board-certified small animal internal medicine diplomates and systemic disease and toxicity were excluded based on clinical history, physical exam, complete blood count, serum biochemistry, urinalysis, abdominal imaging. Exclusion criteria included the administration of non-steroidal anti-inflammatory drugs, glucocorticosteroids, or any antibiotic in the week prior to presentation; any systemic disorder not consistent with AHDS or its complications; the presence of parasites in the feces (via fecal floatation); or positive test results for canine parvovirus using a commercially available ELISA test (IDEXX, Westbrook, Maine). Dogs were excluded if the histopathologic changes in the colonic biopsies were not consistent with AHDS [5, 8, 13].

## Study design

This study was a randomized, placebo-controlled, open-label, longitudinal, prospective trial (S1 Fig). Dogs with AHDS were prospectively recruited following signed owner consent and randomized to the sham-treatment control group or the FMT treatment group according to a premade randomization table generated from an online randomization tool (https://www.randomizer.org). Feces were collected with a fecal loop on admission, at discharge from the clinic, and 1-month post-discharge. In contrast, feces from donor dogs were collected immediately after natural defecation. For all of the study's time points, a portion of the donors and AHDS dogs' stool (1 mL) was mixed with 1 mL RNAlater (Invitrogen ThermoFisher Scientific, Waltham, MA) inside a cryovial (Simport, Beloeil QC, Canada) immediately after collection of the stool for 16S-rRNA sequencing. Immediately after group allocation symptomatic treatment (intravenous fluids) was initiated as needed, and a colon cleansing protocol (below) was commenced in all dogs. All patients underwent a colonoscopy (below) with biopsies taken for histopathologic evaluation. After collection of the biopsies, either FMT solution or a similar volume of saline was instilled within the proximal colon adjacent to the cecum in accordance to dogs group allocation. Supportive medical treatment was identical between groups. The decision to administer treatment via colonoscopy at the proximal colon was based on the authors' opinion that the administered material would be delivered into the colon in the most controlled manner and would stay longer in the colon before it is eliminated by the dogs. Also, it allowed the authors to obtain biopsies that were used to ascertain if the histologic changes in the colon are consistent with those previously reported for AHDS [5, 8, 13]. The nature of the study that involved 2 institutions necessitated utilizing fresh stool from different donors. The authors also think that using stool from multiple donors better simulates the variability in the composition of bacterial communities among the donors' stool in a real-life FMT scenario.

## AHDS clinical score

Once daily, an AHDS clinical score was calculated for each of the hospitalized dogs. The score (S1 Table) was adapted from a study by Mortier et al. [7]. Briefly, the dogs were scored from 0–3 for activity, appetite, vomiting, fecal consistency, and the number of bowel movements, with the highest score possible being 15 correlating with most severe clinical presentation [7].

## Donor screening

Fecal donors were selected if they met the following criteria: healthy (i.e., normal appetite, no vomiting, no diarrhea, no polyuria or polydipsia, no cough, no chronic conditions or any other concerns noticed in the 4 weeks preceding sample collection), vaccinated yearly (canine distemper virus, canine adenovirus II, canine parvovirus, canine parainfluenza), dewormed

every six months, negative for parasitic eggs on fecal floatation, without a history of vomiting, diarrhea or treatment with antibiotics for at least 3 months prior to fecal collection for FMT.

### Preparation of feces for FMT

Freshly passed donor stool was collected immediately after natural elimination on the morning of treatment, placed inside a Ziplock bag or a twist-top container and chilled on ice, and used within 6 hours of collection. Stool was homogenized at room temperature in a sterilized blender at a ratio of 1-part stool/4 parts saline. The suspension was passed through a sterilized sieve to remove large particles. Evidence suggests that when human donor stool is used up to 6 hours from defecation, it is still clinically effective in increasing the diversity in intestine of RCI recipient patients despite a progressive loss of anaerobic bacteria due to limited exposure to oxygen during stool processing [29, 30].

### Colon preparation protocol

A nasoesophageal tube was placed in all AHDS dogs and secured. All colon preparation medications were administered via the nasoesophageal tube in all AHDS dogs. At time 0, 20 mg bisacodyl (Dulcolax®, Sanofi, Bridgewater, NJ) mixed in 4 mL/kg lukewarm water was administered. At times 1-hour and 5-hour, 1 packet of polyethylene glycol (Klean Prep®, Helsinn-Birex Pharmaceuticals Ltd, County Dublin, Ireland) dissolved in 4 mL/kg of lukewarm water was administered. For times 2–4 and 6-8-hours, 4 mL/kg lukewarm water was administered every hour.

### Colonoscopy and fecal transplantation procedure

Colonoscopy and fecal transplantation for all AHDS dogs was performed under general anesthesia approximately 18–24 hours after the end of the colon preparation protocol. Fecal transplantation or sham-treatment with saline was performed at the end of the colonoscopy and biopsy procedure. Approximately 10–15 mL/kg of filtered feces solution or saline was instilled within the proximal colon (ascending colon) next to the cecum, and the endoscope was then removed.

### Periprocedural handling

Before the colonoscopy, the dogs were fasted and only received medication according to the colon prep protocol. Post-procedure they were fed a combination of boiled chicken and Hills I/D tinned diet. However, 2 dogs (1 sham control and 1 FMT recipient) were fed Royal Canin GI in hospital and at discharge.

### Preservation of feces for analysis

One mL of feces from both AHDS and donor dogs at each of the fecal collection time points (i.e., admission, discharge and 30 days after discharge), was immediately mixed inside a cryovial with 1 mL of RNAlater added and then stored at -80˚C for 16S-rRNA amplicon sequencing.

### Histopathology

All samples obtained during colonoscopy were submitted in 10% neutral buffered formalin to a veterinary histology laboratory that processed the formalin-fixed endoscopic biopsies, embedded them in paraffin wax, cut 3-μm thick histological sections, and stained each with

hematoxylin and eosin. A board-certified veterinary pathologist reviewed all slides and was blinded as to group allocation.

## 16S-rRNA amplicon sequencing

Fecal genomic DNA was extracted using the Bioline II Fecal DNA kit (Bioline, London, UK), according to the manufacturer's protocol. DNA extracts were diluted 1:10 with molecular grade water and 20 μL was analyzed on a 1% agarose gel along with a high molecular weight marker (ThermoFisher Scientific, Waltham, MA) to assess DNA quality. Samples were quantified by Qubit HS assay (ThermoFisher Scientific, Waltham, MA) and normalized to 5 ng/μL with Nuclease-Free Water (ThermoFisher Scientific, Waltham, MA). For the amplicon PCR, 1 μL of each normalized DNA was added to 17 μL of AccuPrime Pfx SuperMix (ThermoFisher Scientific, Waltham, MA) and 1 μL each of the barcoded forward (16Sf V3) and reverse (16Sr V4) V3-4 amplicon primers (S2 Table). The following thermocycler program was run: 95˚ C for 2 minutes, then 30 cycles of 95˚ C for 20 seconds, 55˚ C for 15 seconds, 72˚ C for 5 minutes, followed by a final extension cycle 72˚ C for 10 minutes and hold at 4˚ C. A clean-up was performed on the PCR products using SequalPrep™ Normalization Plate Kit (ThermoFisher Scientific, Waltham, MA). The library concentration was quantified by Qubit HS assay (ThermoFisher Scientific, Waltham, MA). The full library size of ~630 bp was verified on a PerkinElmer LabChip GX Touch HT instrument using the DNA High Sensitivity LabChip Assay. The libraries were pooled by equal molarity. The pooled library was diluted to 2 nM with 10 mM Tris pH 8.5 with 0.1% Tween 20 for sequencing. Ten uL of the 2 nM pooled library was denatured to single strand DNA with 10 μL 0.2N NaOH (pH > 12.5) by mixing and incubating at room temperature for 5 minutes. Illumina PhiX Control v3 (Illumina, San Diego, CA) was diluted to 2 nM with 10 mM Tris pH 8.5 with 0.1% Tween 20 then denatured with 10 μL 0.2N NaOH (pH>12.5) by mixing and incubating at room temperature for 5 minutes. Pooled library was diluted to 8 pM and PhiX was diluted to 12.5 pM with ice cold HT-1. Finally, 800 μL of pooled library and 200 μL of PhiX were combined to give a calculated spike of 20% PhiX. Samples were mixed and 600 μL was loaded into a thawed Illumina MiSeq V2 cartridge for sequencing on the Illumina MiSeq platform using paired-end sequencing with 250 nucleotide read length. The resultant sequences were deposited in the National Center for Biotechnology Information (NCBI) BioProject repository (https://www.ncbi.nlm.nih.gov/bioproject/) under the accession number PRJNA629659.

## Analysis of sequencing data

Processing of the sequence reads was performed wholly within the QIIME2 environment (version 2017.10) as an Ubuntu VirtualBox virtual machine [31, 32]. A manifest file was made for the sequences, and they were imported via the q2-demux plugin into an artefact in the format 'PairedEndFastqManifestPhred33' and as the type 'SampleData[PairedEndSequencesWithQuality]'. The data were summarized to visually check on the importation using the QIIME2 viewer (https://view.qiime2.org/), and to assess the parameters required for subsequent sequence quality control and trimming. The dada2 plugin within QIIME2 (q2-dada2) was used to generate a feature table, as well as representative sequence variants [33]. The trimming parameters used were '—p-trim-left-f', '—p-trim-left-r', '—p-trunc-len-f' and '—p-trunc-len-r' and were set to the values 10, 20, 250 and 250 respectively. The feature table was tabulated and then summarized with a metadata file (via q2-feature-table). The sequence variants were aligned using MAFFT (via q2-alignment), subsequently masked and both a rooted and unrooted phylogeny was generated using fasttree (via q2-phylogeny) [34, 35]. Operational taxonomic unit (OTU) taxonomy was performed using a naïve Bayes taxonomy classifier (via

q2-feature-classifier classify-sklearn) against the Greengenes 13_8 99% OTUs reference sequences [36, 37].

Processing of demultiplexed, trimmed, and aligned sequences was performed using the phyloseq package [38]. The demultiplexed, trimmed, and aligned sequence count matrix contained 7,791,596 sequence reads and 2375 OTUs. The median number of reads per sample was 204,361 (range: 39,274–397,760). The phyloseq package was used to process the sequence count matrix in preparation for statistical analysis [38]. First, non-bacterial OTUs and those with an unassigned phylum were removed. Next, OTUs were filtered to exclude low-abundance using a 5% prevalence threshold. Finally, the filtered count matrix was agglomerated on the taxonomic level of genus. The final filtered, agglomerated count matrix contained 4,770,860 sequences and 87 OTUs. The median number of reads per samples was 140,216 (range: 10,569–241,059).

## Statistical analysis

A priori power sample size analysis indicated that a sample size of 4 dogs per group was sufficient to detect a significant difference in 'day of discharge' from the hospital between groups ($\Delta(\mu1-\mu2) = 2d$, $\sigma = 1d$, alpha probability error of 0.05, power of 0.8; G*Power software version 3.1.9.2).

The statistical analyses of AHDS clinical scores, packed cell volume and total solids were performed with statistical package SAS 9.4 (SAS Institute Inc, Cary, NC). The data were assessed for normal distribution by visual inspection of the data on quantile-quantile plots, histogram plots, and by the Shapiro-Wilk test. The normally distributed data were described by mean and standard deviation while non-normally distributed data were described by median and interquartile ranges (IQR). The nonparametric Wilcoxon two sample test was used to assess differences in AHDS clinical scores between and within groups of dogs with AHDS with regards to time of admission and discharge, and to assess if a difference in AHDS clinical scores between admission to discharge was present between groups. A t-test was used to assess differences in hematocrit and serum total proteins between and within groups of dogs with AHDS with regards to time of admission and discharge, and also to assess if a difference in hematocrit and serum total proteins between admission to discharge was present between groups.

All analyses were conducted in the R language for statistical computing (version 3.6.1) [39]. Results were considered significant if $P < 0.05$. For multiple comparisons in the differential abundance testing, we calculated the false discovery rate (FDR) according to the Benjamini-Hochburg method and limited significant results to those with FDR < 0.2 [40]. Data and R code used for these analyses are available in our GitHub repository (https://github.com/pcbarko/Gal-Barko-K9-AHDS-FMT).

The raw, unfiltered count matrix was used to calculate the Shannon diversity index (SDI), an estimate of alpha diversity, for each sample [41]. The SDI weighs species richness and evenness equally and was previously shown to be a robust index for comparison of microbial alpha diversity [42]. The Welch's 2-sample t-test was used to detect statistical differences in SDI between dogs with AHDS and healthy donors at the time of admission. Mixed-effects generalized linear models, implemented with the lmer4 package, were used to detect statistical differences in SDI among treatment groups and sample collection timepoints [43]. To further characterize donor-recipient relationships, we calculated the number of shared OTUs and the Jensen-Shannon divergence index (JSD) among all donor-recipient pairs. We used a repeated measures ANOVA to compare shared OTUs and JSD indices among the sample collection time points.

Unsupervised methods were used to detect changes in microbiota community structure associated with AHDS and FMT. A heatmap was generated from the filtered, agglomerated count matrix after normalizing to counts per million reads (CPM) and $\log_{10}$ transformation. Beta diversity was estimated using the Bray-Curtis dissimilarity matrix generated from the filtered, agglomerated count matrix and visualized with principal coordinate analysis (PCoA) [44]. We used permutational multivariate analysis of variance (PERMANOVA) to detect statistical differences among the groups of samples visualized with the PCoA plots. PERMANOVA was implemented on the Bray Curtis dissimilarity matrix with 1,000 permutations using the adonis function in the vegan package (38).

To detect differentially abundant OTUs due to diagnosis, treatment group, and collection time, an empirical Bayesian linear model was implemented using the Glimma package [45]. The count matrix was normalized to counts per million and log transformed. The Glimma model included the individual dog ID as a random effect (repeated measures) and treatment group and collection time as fixed effects.

To identify specific subnetworks of OTUs associated with FMT, we constructed a weighted eigenvalue network using the WGCNA package. First, the filtered and agglomerated OTU abundance matrix was normalized using the total sum scaling method and transformed by the centered log-ratio. Next, outliers were detected by examining a cluster dendrogram of samples based on their Euclidean distance and calculating the standardized network connectivity for each sample. To determine which soft thresholding power best approximated a scale-free topology (SFT), we examined plots of the scale-free topology fit index (SFT $R^2$) as a function of soft-thresholding power from 1 to 20. We selected the lowest soft-thresholding power that approximated a scale-free topology (SFT $R^2 > 0.8$). The final weighted co-abundance network was generated using the blockwiseModules function and the following parameters: corType = pearson, power = 4, minimum module size = 5, merging threshold = 0.2, and deepSplit = 2, networkType = signed hybrid. This output included module eigenvalues (MEs), which are the first principal component of each module and thus a mathematical summary of variation each sample contributes to the module. Module eigenvalues for each sample were used with linear mixed-effects models to compare module abundance among treatment groups and collection times. To identify important OTUs within each module, module membership (kME) was determined by correlating the abundance of each OTU with the eigenvalues of each module. Significant OTUs within were those with |kME| greater than 0.5.

## Results

The dogs that received FMT did not differ in median (IQR) AHDS clinical score from the sham-treated control dogs at admission (5.5 (3) vs. 8.5 (4.5); P = 0.22; n = 8) or at discharge from the hospital (3.5 (2.5) vs. 2 (2); P = 0.41; n = 8). While the AHDS clinical score did not differ between groups, the median (IQR) AHDS clinical score of sham-treated control dogs differed between the time of admission and discharge (8.5 (4.5) vs. 2 (2); P = 0.04; n = 4); whereas the median (IQR) AHDS clinical score of FMT treated dogs did not (5.5 (3) vs. 3.5 (2.5); P = 0.22; n = 4). However, the difference in median (IQR) AHDS clinical score between discharge to admission of FMT treated dogs did not differ from that of sham-treated control dogs (-1.5 (3.5) vs. -5.5 (4.5), P = 0.08; n = 4). The mean (±SD) hematocrit of FMT recipient dogs did not differ from sham-treated control dogs on admission (65±1 vs. 67±12, P = 0.85) and discharge (47±14 vs. 51±10, P = 0.71). Similarly, the mean (±SD) total protein of FMT recipient dogs did not differ from sham-treated control dogs on admission (62±2 vs. 76±11, P = 0.1) and discharge (52±7 vs. 49±13, P = 0.76). The mean difference between admission and discharge did not differ between FMT recipient sham-treated control dogs for hematocrit

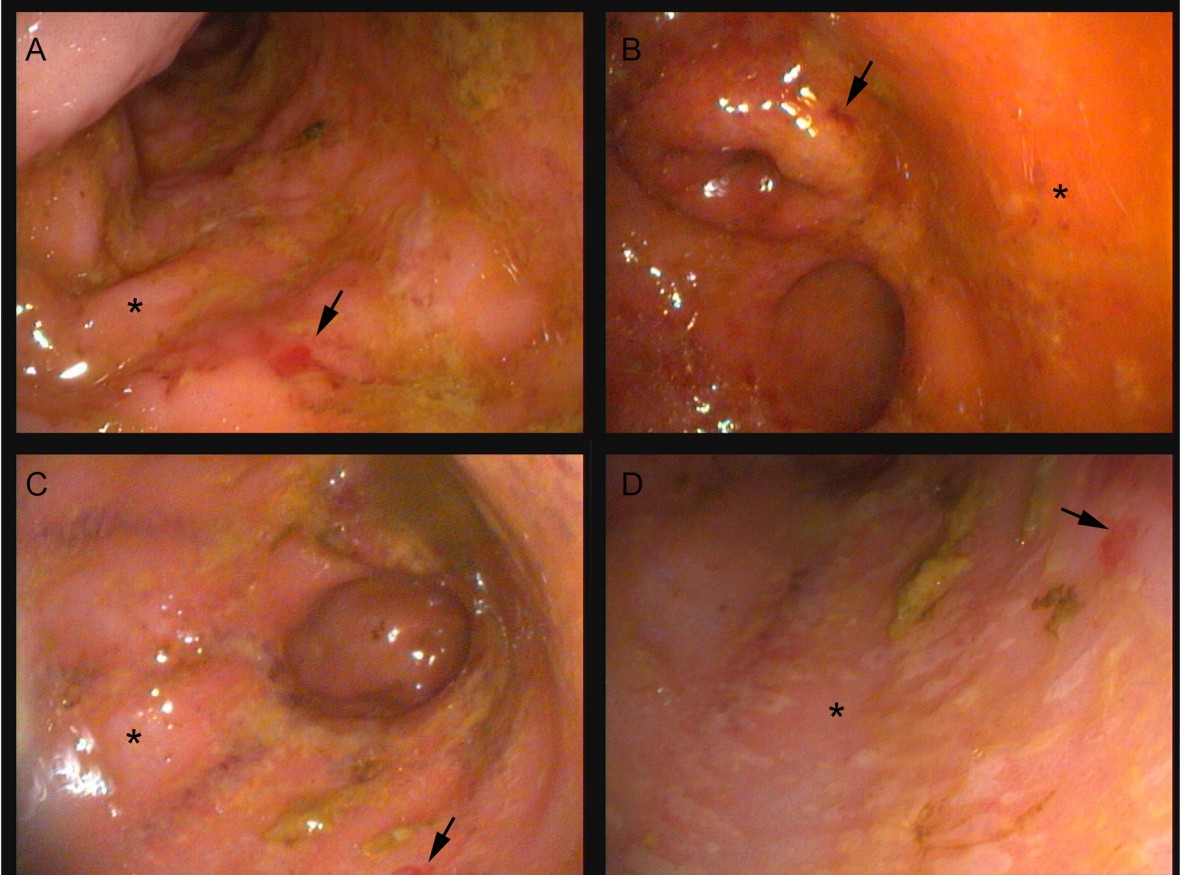

**Fig 1. Colonoscopy images, proximal colon, dogs with AHDS.** (A & B) Sham-treated control dog. (C & D) Fecal microbiota transplantation-recipient dog. The black arrows indicate areas of ulceration and hemorrhage. The black asterisks indicate dull edematous colonic mucosa. AHDS: acute hemorrhagic diarrhea syndrome.

(P = 0.89) and total protein (P = 0.31). Regardless of their group allocation, all the dogs with AHDS had a rapid and similar clinical recovery and left the hospital on the morning of the third day of hospitalization. The owners of all AHDS dogs reported that their dogs were healthy at 30 days post-discharge and the feces of the dogs on 30-day had normal formed consistencies. The donor dogs remained healthy and had normal formed stools throughout the study period (i.e., admission-to-30-day post-discharge).

The colonic mucosa of healthy dogs is smooth, pale pink, glistening, and has prominent submucosal blood vessels. All dogs with AHDS had minimal-to-mild multifocal mucosal hemorrhages and erythematous rough and dull mucosa with inconspicuous submucosal blood vessels (Fig 1, Table 2). These changes were interpreted as consistent with AHDS. The histopathology of all dogs had various degrees (often minimal-to-mild) of mucosal ulceration and necrosis with hemorrhage and neutrophilic inflammation. These were often intervening with areas of mucosal regeneration (Fig 2, Table 2). In addition to the aforementioned histopathologic changes, 2 of the FMT-recipient dogs and 1 of the sham-control dogs also had infiltration of the colonic mucosal lamina propria by plasma cells and eosinophils suggesting a possible background enteropathy.

We contrasted the SDI of healthy donor dogs with that of dogs with AHDS at the time of admission (Fig 3). We found that the dogs with AHDS had a significantly lower mean (±SE)

**Table 2. Endoscopic and histologic changes in dogs with AHDS.**

| Endoscopic changes Severity | Colon | | | |
|---|---|---|---|---|
| | Minimal | Mild | Moderate | Severe |
| Hyperemia | FT(2); C(2) | FT(1); C(1) | FT(1); C(1) | 0 |
| Edema | FT(1); C(1) | FT(1); C(3) | FT(2) | 0 |
| Hemorrhage | FT(3); C(1) | FT(1); C(3) | 0 | 0 |
| Erosion/ulcers | FT(3); C(3) | 0 | FT (1); C(1) | 0 |
| **Histologic changes Severity** | **Minimal** | **Mild** | **Moderate** | **Severe** |
| Epith. Injury | FT(3); C(2) | FT(1); C(2) | 0 | 0 |
| L. p. neutrophils | FT(2); C(1) | FT(2); C(3) | 0 | 0 |
| L. p. hem | 0 | FT(4); C(4) | 0 | 0 |
| Epith. reg | 0 | FT(4); C(4) | 0 | 0 |
| Other | 0 | $FT_{Eo+PC}(1)$; $C_{Eo+PC}(1)$ | $FT_{Eo+PC}(1)$ | 0 |

C: sham-control followed by numbers of cases affected by change; FT: fecal microbial transplantation recipient followed by numbers of cases affected by change; Epith. Injury: epithelial injury; L. p. Neutrophils: lamina propria neutrophils; L. p. Hem: lamina propria hemorrhage; Epith. Reg: epithelial regeneration; Eo+PC: infiltration of the colonic lamina propria by plasma cells and eosinophils.

SDI at presentation (2.82±0.2) compared to healthy donor dogs (3.65±0.2; P = 0.002; Fig 3A). At admission, the mean (±SE) SDI of dogs with AHDS that received FMT (3.09±0.263) did not differ significantly from the sham-treated control dogs (2.55±0.263; P = 0.053; Fig 3C). We then determined the temporal effect of FMT on the microbiota diversity of FMT recipients and sham-treated control dogs with AHDS. We found that the mean (±SE) SDI of dogs with AHDS that received FMT significantly increased after FMT (admission, 3.09±0.19 vs. discharge, 3.72±0.19; P = 0.04; Fig 3B) and their SDI did not differ from the donor dogs (FMT recipient, 3.72±0.19 vs. donors at discharge, 3.46±0.19; P = 0.33, Fig 3C). In contrast, at discharge, the mean (±SE) SDI of the FMT recipients (3.72±0.19) and donor dogs (3.46±0.19) significantly differed from the mean (±SE) SDI of the sham-treated control dogs with AHDS (2.71±0.19; P = 0.002 and P = 0.01; Fig 3C). There were no significant differences in mean (±SE) SDI between FMT recipients (3.09±0.263) and sham-treated control dogs (3.09 ±0.263) with AHDS at 30 days (P = 0.87; Fig 3C). The mean (±SE) SDI of sham-treated control dogs with AHDS did not significantly change over time (admission, 2.55±0.19 vs. discharge, 2.71 ±0.19 vs. 30 days, 3.09±0.19; P>0.05; Fig 3B).

Principal coordinate analysis of the Bray Curtis dissimilarity matrix separated dogs with AHDS from healthy dogs along principal coordinate 1 (Fig 4A). The observed separation of samples by diagnosis (AHDS vs. healthy donors) was significant (PERMANOVA P = 0.003). At discharge and 30 days, dogs with AHDS that received FMT were separated from their corresponding pre-FMT (admission) samples and sham-treated control dogs with AHDS along principal coordinate 1 (Fig 4B). The observed separation of samples by treatment group (FMT vs. control; PERMANOVA P = 0.009) and collection time (admission vs. discharge vs. 30 days; PERMANOVA P = 0.049) was significant.

The number of OTUs shared between donor-recipient pairs increased following FMT in all dogs and remained increased at 30 days in 3 out of 4 dogs; however, the difference was not significant (Fig 5A; ANOVA P = 0.088). Divergence between donor-recipient pairs decreased significantly after FMT (Fig 5B; ANOVA P = 0.004) at both discharge (P = 0.047) and 30 days (P = 0.047), compared with admission.

To test for changes in abundance of OTUs associated with FMT, we used an empirical Bayesian linear model. Twenty-eight taxa were found to vary significantly in abundance across

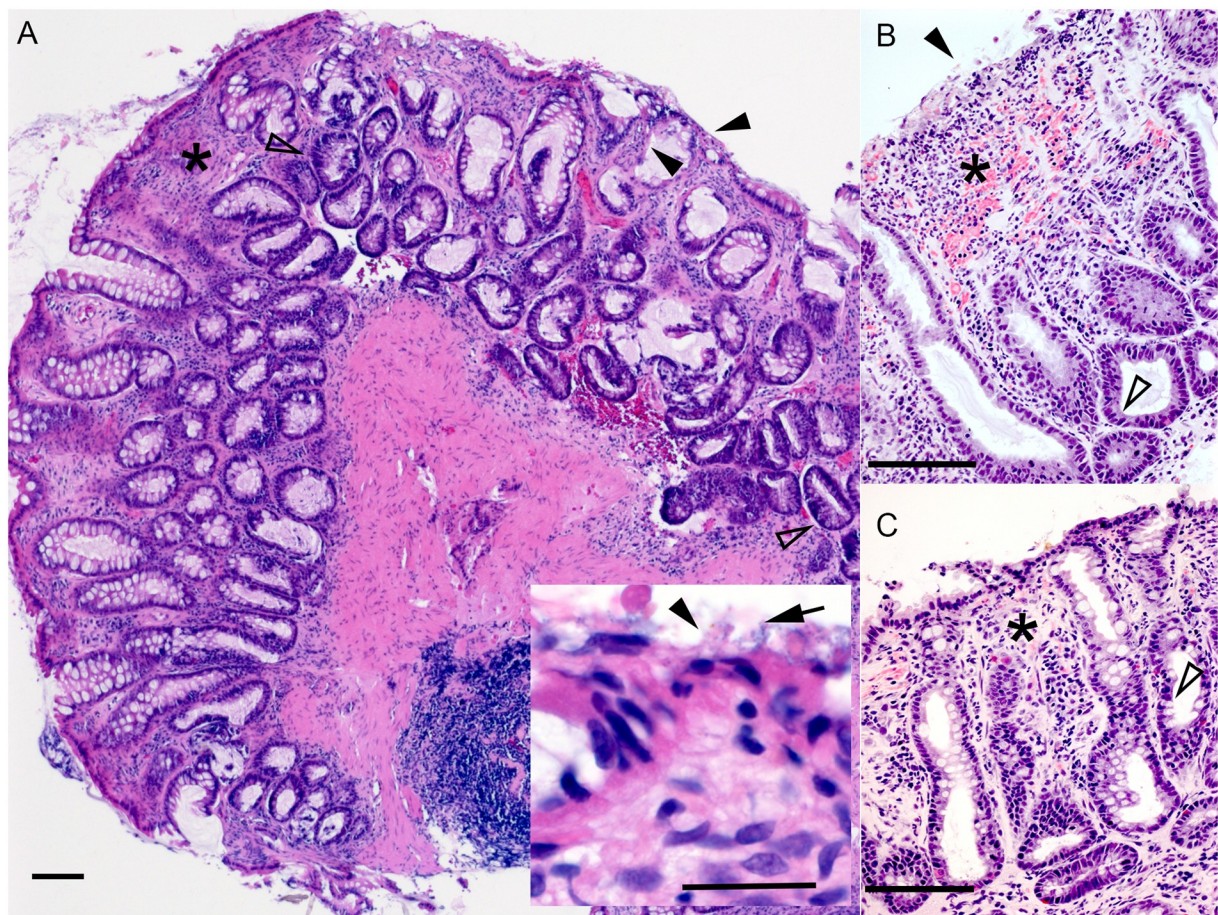

**Fig 2. Colonic histopathology, proximal colon, dogs with AHDS.** (A) FMT-recipient dog from New Zealand. H&E, bar = 100 μm. Inset, H&E, bar = 25μm. (B & C) FMT-recipient and sham-control dogs from South Africa. H&E, bar = 100 μm. Black arrowhead: denuded ulcerated colonic mucosa; Black arrow: denuded mucosa with adherent colonic bacteria; Black asterisk: leukocytes, admixed with hemorrhage and edema infiltrate the colonic lamina propria. The latter has loss of colonic glands with collapse of the supporting stroma; Open arrowhead: 1–2 layers of large hyperchromatic colonic epithelial cells pile up and exhibit frequent mitoses (regeneration).

treatment groups and collection times (Fig 6A). A heatmap of the OTU abundance demonstrates the broad shift in these communities in FMT recipients (Fig 6B). Compared with donor dogs at admission, those with AHDS had a significantly lower abundance of 15 OTUs and a higher abundance of 1 OTU (Fig 6A; Table 3). Compared with sham-treated control dogs, FMT recipient dogs had 18 OTUs that were significantly increased at discharge (Fig 6A; Table 4), and 4 OTUs that were significantly increased at 30 days (Fig 6A; Table 5). No OTUs were significantly different between donor dogs and FMT recipients at 30 days. In contrast with healthy donors, sham-treated controls had significantly lower abundances of *Slackia*, *Prevotella copri*, *Megamonas*, *Faecalibacterium*, *Catenibacterium*, *Butyricicoccus pullicaecorum*, *Anaerotruncus*, *Eubacteria biforme* at discharge (Table 4), and *Slackia*, *Prevotella copri*, *Faecalibacterium prausnitzii*, *Catenibacterium*, *Eubacteria biforme* at 30 days (Table 5).

Many of the OTUs that were significantly decreased in AHDS dogs on admission and increased in FMT recipients are known to have metabolic functions that are important to host animal health (Fig 7) [46]. Short-chain fatty acid (SCFA) producers including *Eubacterium biforme*, *Faecalibacterium prausnitzii*, and *Prevotella copri* were significantly decreased in AHDS dogs and increased in FMT-recipients compared with sham-treated controls (Fig 7A–

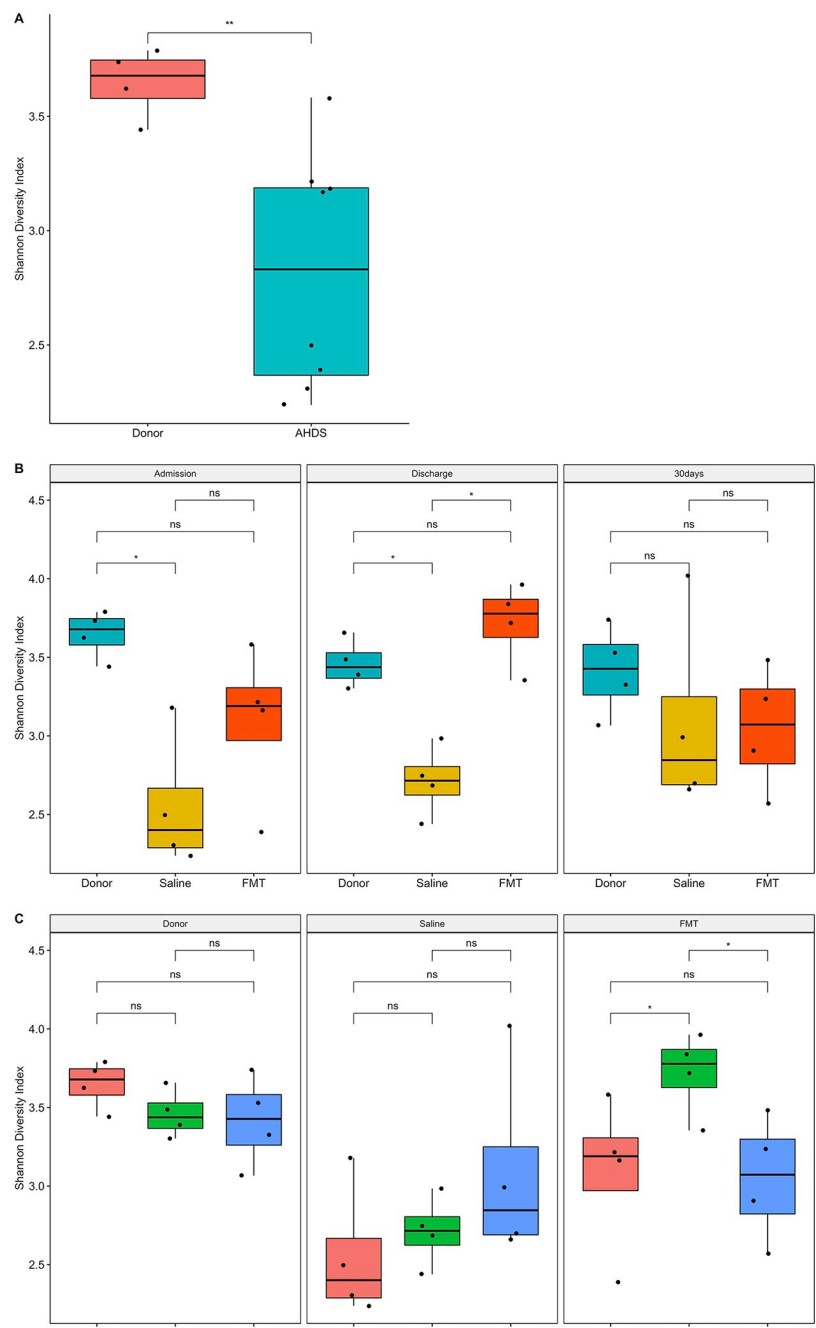

**Fig 3. Boxplot of Shannon diversity indices demonstrates increased microbiota diversity in FMT-recipients.** The Shannon diversity index (SDI) was used to quantify microbiota alpha diversity among groups of samples. (A) SDI of dogs with AHDS compared with healthy donor dogs at the time of admission (pre-FMT). (B) Within group comparisons of SDI in FMT recipient, sham-treated control dogs, and donor dogs throughout the study. (C) Between group comparisons of SDI in FMT recipient, sham-treated control dogs, and donor dogs throughout the study. Blue circles: South Africa dogs; Gray circles: New Zealand dogs; AHDS: acute hemorrhagic diarrhea syndrome; FMT: fecal microbial transplantation. The red X are the median and vertical red bars are the interquartile range. The colored circles are individual dogs; the blue circles are dogs from South Africa, and the grey circles are dogs from New Zealand. Statistical comparisons among treatment groups and collection times are indicated by annotated brackets where "ns" represents P $\geq$ 0.05, "*" represents 0.01 < P < 0.05, and "**" represents 0.001 < P < 0.01.

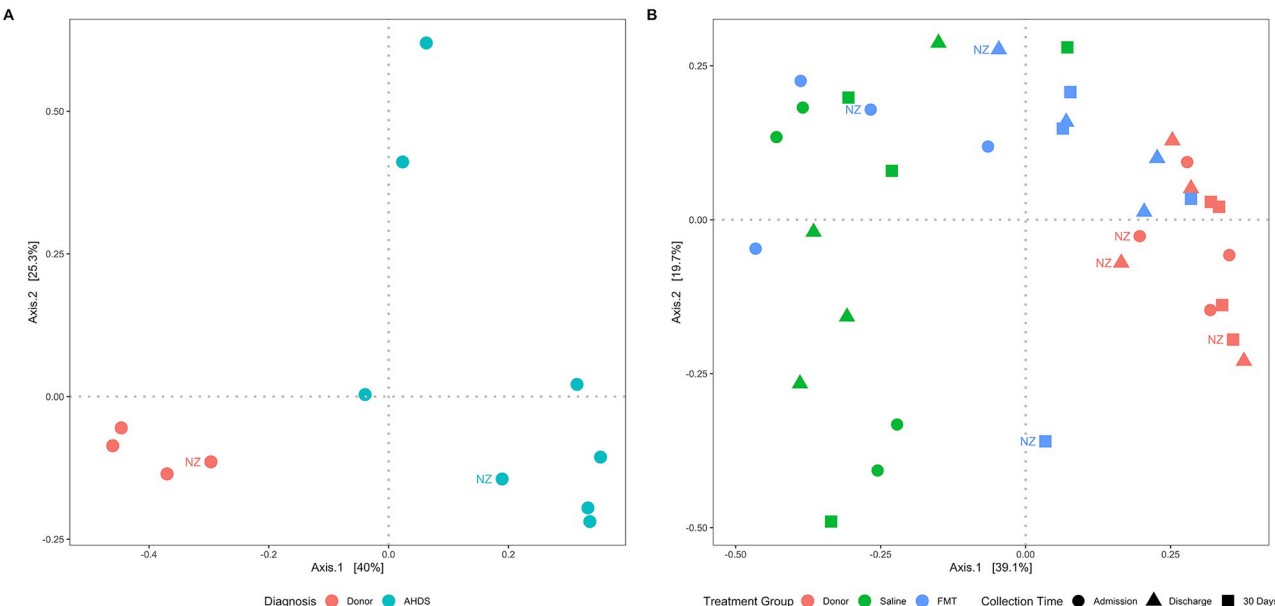

**Fig 4. Principal coordinate analysis of the Bray Curtis dissimilarity matrix reveals changes in community composition associated with FMT.** The Bray-Curtis dissimilarity was visualized using principal coordinate analysis (PCoA) and compared using permutational analysis of variance (PERMANOVA). (A) Dogs with AHDS separate significantly (P = 0.003) from healthy controls at the time of hospital admission (pre-FMT) along the first principal coordinate (x-axis). (B) Dogs treated with FMT separate from sham-treated controls and pre-treatment samples (circles) at discharge (triangles) and 30 days (squares) and cluster closer to the donor dogs along the first principal coordinate. Separation of groups based on treatment group (P = 0.003) and collection time (P = 0.049) were significant. AHDS: acute hemorrhagic diarrhea syndrome; FMT: fecal microbial transplantation; NZ: New Zealand dogs.

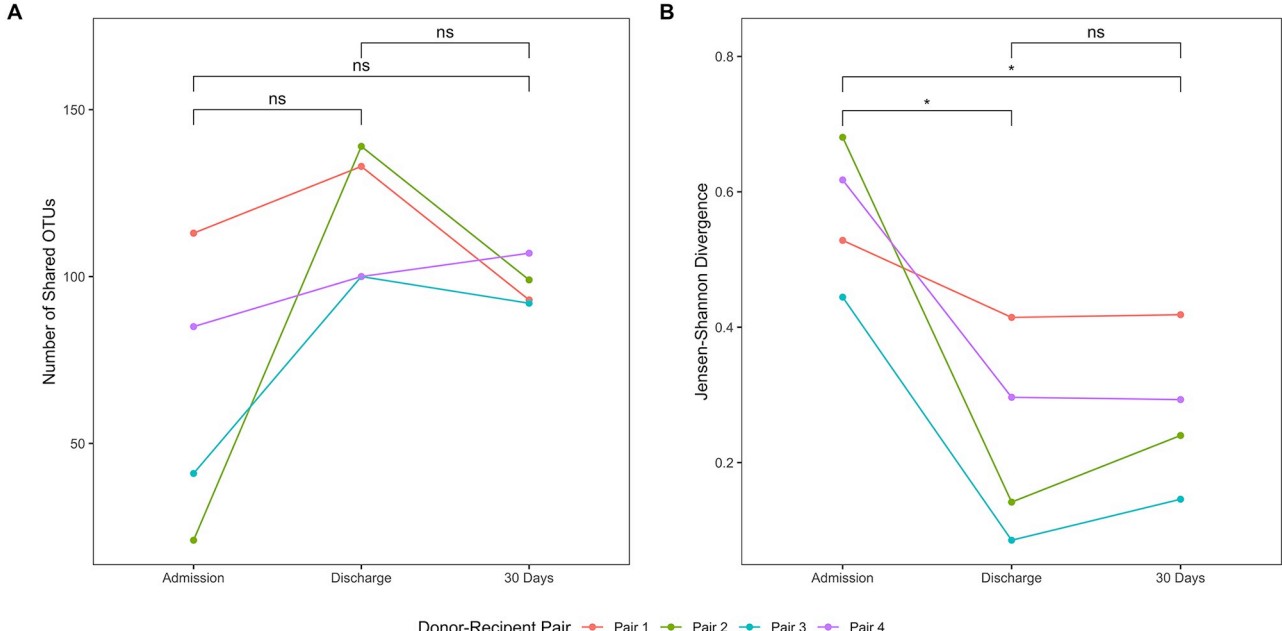

**Fig 5. Analysis of relationships among donor-recipient pairs.** The number of shared OTUs and Jensen-Shannon divergence were used to characterize associations between donor-recipient pairs. Results were compared using a repeated measures ANOVA. (A) The number of OTUs shared among donor-recipient pairs increased following fecal microbial transplantation, but the results were not significant. (B) Divergence among donor-recipient pairs decreased significantly from admission to discharge, and from admission to 30 days. Statistical comparisons among treatment groups and collection times are indicated by annotated brackets where "ns" represents P ≥ 0.05 and "*" represents 0.01 < P < 0.05.

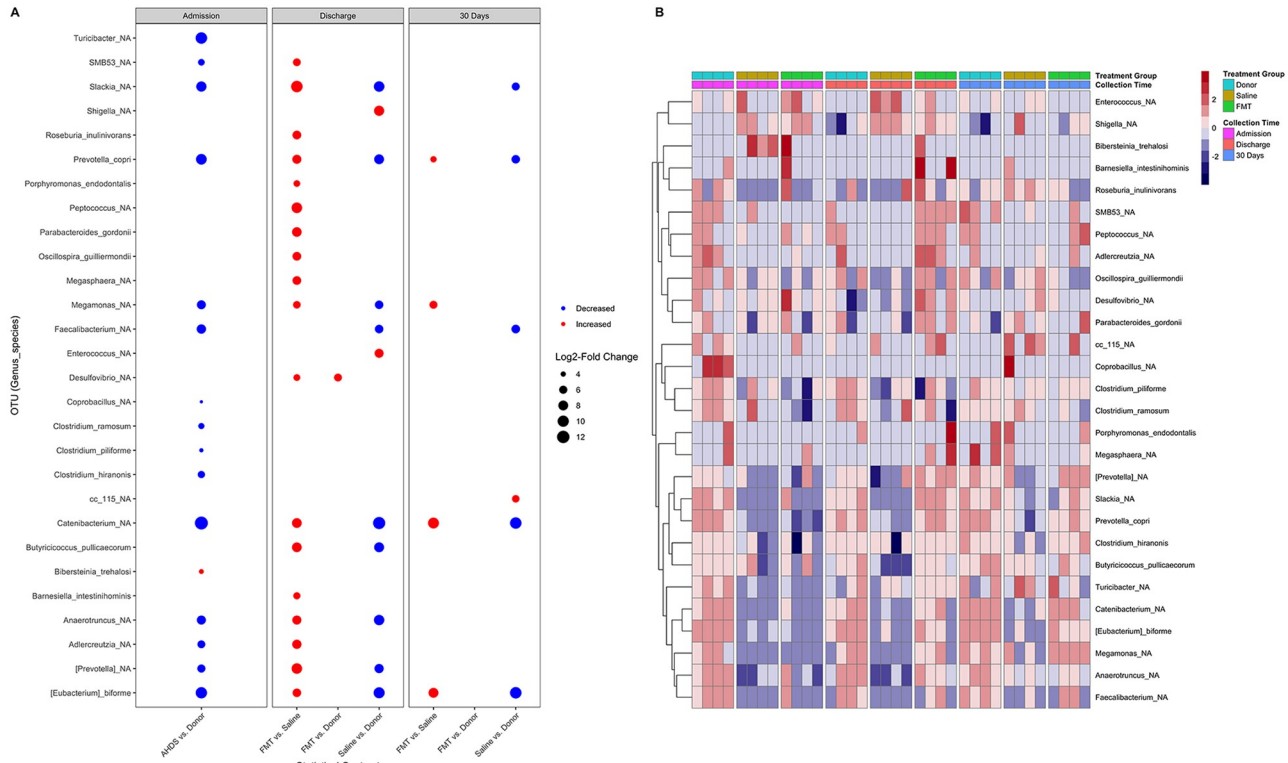

**Fig 6. Differentially abundant taxa associated with FMT.** (A) An empirical Bayesian linear model was used to detect differentially abundant taxa associated with AHDS and FMT. Taxa are present on the y-axis and the specified statistical comparisons are represented on the x-axis. The color of each point corresponds to whether the taxon was increased (red) or decreased (blue). The size of each point is proportional to the log2 of the fold-change between the specified groups. (B) Heatmap of the OTU abundance demonstrating broad shifts in communities of FMT recipients AHDS: acute hemorrhagic diarrhea syndrome; FMT: fecal microbial transplantation.

7G). The abundance of *Clostridium hiranonis*, a species that converts primary bile acids into secondary bile acids, was decreased in AHDS dogs at admission and increased in FMT recipients compared with sham-treated controls, though this change was not significant (Fig 7H).

Seven co-abundance subnetwork modules, represented by 7 unique colors, were generated from the OTU abundance data (Fig 8A). Of these, one module (colored brown in Fig 8B) contained many of the same OTUs (see above) found to be differentially abundant in dogs that received FMT. Furthermore, the abundance in that module was increased in FMT recipients at discharge and 30 days, compared with dogs that received sham treatment with saline (Fig 8C and 8D).

## Discussion

Acute hemorrhagic diarrhea syndrome is a dramatic disease that is characterized by peracute onset of severe vomiting and watery bloody diarrhea. In two large retrospective studies published in 1977 and 2015, the disease prevalence was 0.51% and 0.87% of the referral hospital population (24,104 and 12,360 dogs, respectively), with reported mortality rates of 10% and 2.7%, respectively [1, 7]. With supportive care, most dogs significantly improve within 48 hours, with a median hospitalization time of 3 days [7, 47]. The rapid resolution of clinical signs of gastrointestinal dysfunction in the dogs with AHDS from our study is thus consistent with those previously reported [7, 47] and could explain why, in the setting of a small sample size pilot study, there were no differences in clinical outcomes between FMT recipients and

**Table 3. Significant changes in taxa at admission (AHDS vs donors).**

| Taxa | Log2FC | *P*-Value | FDR |
|---|---|---|---|
| *Catenibacterium* | -13.04 | 1.98E-07 | 8.59E-06 |
| *Turicibacter* | -10.44 | 8.86E-04 | 9.64E-03 |
| *Eubacterium biforme* | -10.16 | 1.90E-06 | 5.50E-05 |
| *Prevotella copri* | -9.17 | 6.70E-08 | 5.83E-06 |
| *Slackia* | -8.52 | 2.93E-05 | 5.10E-04 |
| *Faecalibacterium prausnitzii* | -7.35 | 1.63E-03 | 1.57E-02 |
| *Anaerotruncus* | -7.01 | 2.68E-05 | 5.10E-04 |
| *Megamonas* | -6.88 | 2.89E-04 | 4.19E-03 |
| *Prevotella* | -5.99 | 3.05E-02 | 1.81E-01 |
| *Adlercreutzia* | -5.74 | 7.58E-03 | 6.59E-02 |
| *Clostridium hiranonis* | -5.21 | 3.32E-02 | 1.81E-01 |
| *SMB53* | -4.76 | 2.62E-02 | 1.76E-01 |
| *Clostridium ramosum* | -4.42 | 2.18E-02 | 1.58E-01 |
| *Clostridium piliforme* | -3.68 | 3.31E-02 | 1.81E-01 |
| *Coprobacillus* | -3.58 | 5.10E-04 | 6.34E-03 |
| *Bibersteinia trehalosi* | 3.86 | 1.83E-02 | 1.45E-01 |

AHDS: acute hemorrhagic diarrhea syndrome; Donor: healthy dogs that donated feces for fecal microbiota transplantation; FDR: False discovery rate. An estimate of the proportion of falsely positive findings was generated using the Benjamini-Hochburg method; Log2FC: log2-transformed fold-change in taxa abundance between dogs with AHDS and healthy donors at the time of hospital admission. A positive Log2FC means the taxa is increased and a negative value means the taxa was decreased in the AHDS group compared with the healthy donors; P-Value: P-values generated from a generalized linear model based on a negative binomial distribution.

sham-treated control groups. Additionally, the minimal to mild endoscopic and histologic scores in all AHDS dogs in our study could suggest that at the time of admission, the dogs were already in an early phase of recovery from AHDS, thus making it hard to ascertain treatment-based clinical outcomes due to the small sample size of the study.

However, a striking finding in this study was the marked difference in microbiota diversity between healthy dogs and those with AHDS (Figs 3A and 4A; Table 3). Acute hemorrhagic diarrhea syndrome is an acute necrotizing enterocolitis that has been associated in some studies with *C. perfringens* overgrowth and release of pore-forming netE/F enterotoxins [5, 6, 9, 11, 13, 48]. In the present study, we did not find an increase in the abundance of *C. perfringens* in any of the AHDS dogs. While this is in contrast to several previous studies [5, 6, 9, 11, 13, 48] it is also consistent with other studies in which *C. perfringens* or its toxins were not increased [2, 4, 47]. This highlights the syndromic nature of AHDS in which dogs share common clinical signs and clinicopathologic abnormalities but not necessarily a common etiology. The histopathological changes in the intestine of dogs with AHDS resemble the histopathological lesions in the intestine of people with RCI [13, 49] and could support a pathomechanistic hypothesis of toxin-mediated mucosal injury in AHDS. Aside from this similarity, the two diseases differ with respects to etiology, chronicity, and known predispositions. In the present study, the authors postulated that FMT could accelerate clinical recovery of dogs with AHDS if toxin-mediated mucosal injury is central to the evolution of clinical disease. In that context, previous studies demonstrated significant decreases in the fecal microbial diversity in a mouse model of RCI and in human patients with RCI [14–17, 50] that increase following fecal microbial transplantation [15, 16, 50, 51] and is temporally associated with resolution of the disease [15, 20, 52]. Little is known about the fecal microbial composition and diversity in dogs with AHDS.

**Table 4. Significant changes in taxa at discharge.**

| Taxa | Comparison | Log2FC | P-Value | FDR |
|---|---|---|---|---|
| *Desulfovibrio* | FMT vs. Donor | 5.75 | 1.23E-03 | 1.07E-01 |
| *Porphyromonas endodontalis* | FMT vs. Control | 4.77 | 2.21E-02 | 1.15E-01 |
| *Desulfovibrio* | FMT vs. Control | 4.86 | 5.15E-03 | 5.49E-02 |
| *Barnesiella intestinihominis* | FMT vs. Control | 5.12 | 7.43E-03 | 5.88E-02 |
| *Megamonas* | FMT vs. Control | 5.47 | 8.44E-03 | 6.12E-02 |
| *SMB53* | FMT vs. Control | 5.65 | 2.25E-02 | 1.15E-01 |
| *Eubacterium biforme* | FMT vs. Control | 6.42 | 3.10E-03 | 4.49E-02 |
| *Megasphaera* | FMT vs. Control | 6.58 | 1.80E-02 | 1.09E-01 |
| *Oscillospira guilliermondii* | FMT vs. Control | 6.61 | 1.02E-02 | 6.84E-02 |
| *Roseburia inulinivorans* | FMT vs. Control | 6.64 | 2.72E-02 | 1.31E-01 |
| *Prevotella copri* | FMT vs. Control | 6.99 | 5.70E-05 | 2.48E-03 |
| *Anaerotruncus* | FMT vs. Control | 7.02 | 1.70E-04 | 4.94E-03 |
| *Adlercreutzia* | FMT vs. Control | 7.26 | 3.81E-03 | 4.74E-02 |
| *Parabacteroides gordonii* | FMT vs. Control | 7.69 | 6.27E-03 | 5.49E-02 |
| *Butyricicoccus pullicaecorum* | FMT vs. Control | 7.83 | 1.09E-03 | 2.38E-02 |
| *Catenibacterium* | FMT vs. Control | 7.89 | 1.45E-03 | 2.52E-02 |
| *Prevotella* | FMT vs. Control | 8.94 | 6.31E-03 | 5.49E-02 |
| *Peptococcus* | FMT vs. Control | 9.06 | 1.88E-02 | 1.09E-01 |
| *Slackia* | FMT vs. Control | 10.42 | 1.29E-05 | 1.12E-03 |
| *Catenibacterium* | Control vs. Donor | -11.58 | 1.51E-05 | 4.38E-04 |
| *Eubacterium biforme* | Control vs. Donor | -9.69 | 3.44E-05 | 7.49E-04 |
| *Slackia* | Control vs. Donor | -8.93 | 1.05E-04 | 1.82E-03 |
| *Anaerotruncus* | Control vs. Donor | -8.71 | 9.33E-06 | 4.06E-04 |
| *Butyricicoccus pullicaecorum* | Control vs. Donor | -8.08 | 8.04E-04 | 1.17E-02 |
| *Prevotella copri* | Control vs. Donor | -7.98 | 8.81E-06 | 4.06E-04 |
| *Prevotella* | Control vs. Donor | -7.19 | 2.48E-02 | 1.96E-01 |
| *Faecalibacterium prausnitzii* | Control vs. Donor | -6.4 | 1.40E-02 | 1.22E-01 |
| *Megamonas* | Control vs. Donor | -6.36 | 2.61E-03 | 2.84E-02 |
| *Enterococcus* | Control vs. Donor | 6.96 | 4.49E-03 | 4.34E-02 |
| *Shigella* | Control vs. Donor | 8.39 | 1.77E-03 | 2.20E-02 |

AHDS: acute hemorrhagic diarrhea syndrome; Control: sham-treated control dogs with AHDS; Donor: healthy dogs that donated feces for FMT; FDR: False discovery rate. An estimate of the proportion of falsely positive findings was generated using the Benjamini-Hochburg method; FMT: dogs with AHDS that received fecal microbiota transplantation; Log2FC: log2-transformed fold-change in taxa abundance from different statistical comparisons among groups. A positive Log2FC means the taxa is increased and a negative value means the taxa was decreased in the first vs. the second groups listed in the "Comparisons" column. P-Value: P-values generated from a generalized linear model based on a negative binomial distribution.

In a previous study, dogs with AHDS had significant decreases in SCFA-producing bacteria (*Blautia*, *Ruminococcaceae* including *Faecalibacterium*, and *Turicibacter*), and significant increases in genus *Sutterella* and *Clostridium perfringens* [10, 12]. However, in our study the same taxa did not differ between AHDS dogs and the healthy donors, whereas other SCFA-producing bacteria that were in low abundances in the present study were not described in the previous study. These differences between the studies emphasize the heterogeneity in the composition of gut microbial communities among dogs with AHDS which differ from each other with respect to their genetic background, immunological status, nutrition, and environmental factors, all of which are likely to be important factors that participate in the predisposition to this syndrome. Our study demonstrated that spontaneous clinical recovery in dogs with

**Table 5. Significant changes in taxa at 30 days.**

| Taxa | Comparison | Log2FC | *P*-Value | FDR |
|---|---|---|---|---|
| *Prevotella copri* | FMT vs. Control | 4.54 | 4.89E-03 | 1.12E-01 |
| *Megamonas* | FMT vs. Control | 5.85 | 5.17E-03 | 1.12E-01 |
| *Eubacterium biforme* | FMT vs. Control | 8.04 | 3.48E-04 | 1.51E-02 |
| *Catenibacterium* | FMT vs. Control | 9.52 | 2.01E-04 | 1.51E-02 |
| *Eubacterium biforme* | Control vs. Donor | -10.41 | 1.24E-05 | 1.08E-03 |
| *Catenibacterium* | Control vs. Donor | -10.38 | 6.81E-05 | 2.96E-03 |
| *Faecalibacterium* | Control vs. Donor | -6.52 | 1.24E-02 | 1.80E-01 |
| *Prevotella copri* | Control vs. Donor | -6.39 | 1.76E-04 | 5.12E-03 |
| *Slackia* | Control vs. Donor | -5.86 | 6.49E-03 | 1.29E-01 |
| *cc_115* | Control vs. Donor | 5.44 | 7.44E-03 | 1.29E-01 |

AHDS: acute hemorrhagic diarrhea syndrome; Control: sham-treated control dogs with AHDS; Donor: healthy dogs that donated feces for FMT; FDR: False discovery rate. An estimate of the proportion of falsely positive findings was generated using the Benjamini-Hochburg method; FMT: dogs with AHDS that received fecal microbiota transplantation; Log2FC: log2-transformed fold-change in taxa abundance from different statistical comparisons among groups. A positive Log2FC means the taxa is increased and a negative value means the taxa was decreased in the first vs. the second groups listed in the "Comparisons" column. *P*-Value: *P*-values generated from a generalized linear model based on a negative binomial distribution.

AHDS is not associated with significant changes in alpha and beta diversity (Figs 3B and 4B), and compared to healthy donor dogs, the microbiome of sham-treated control AHDS dogs was associated with a persistently decreased abundance of the SCFA-producing bacteria *E. biforme*, *Catenibacterium*, *Faecalibacterium*, and *Prevotella copri* at 30 days post-discharge (Figs 6A and 7; Table 5).

In marked contrast to the spontaneously recovering sham-treated control AHDS dogs, FMT recipients had a significant increase in SDI at discharge (Fig 3B). This was accompanied by increases in the abundance of SCFA-producing bacteria including *Prevotella* copri, *Roseburia inulinivorans*, *Butyricicoccus pullicaecorum*, and *Eubacterium biforme* (Table 4). While there were no significant differences in SDI among groups at 30 days, a single FMT procedure

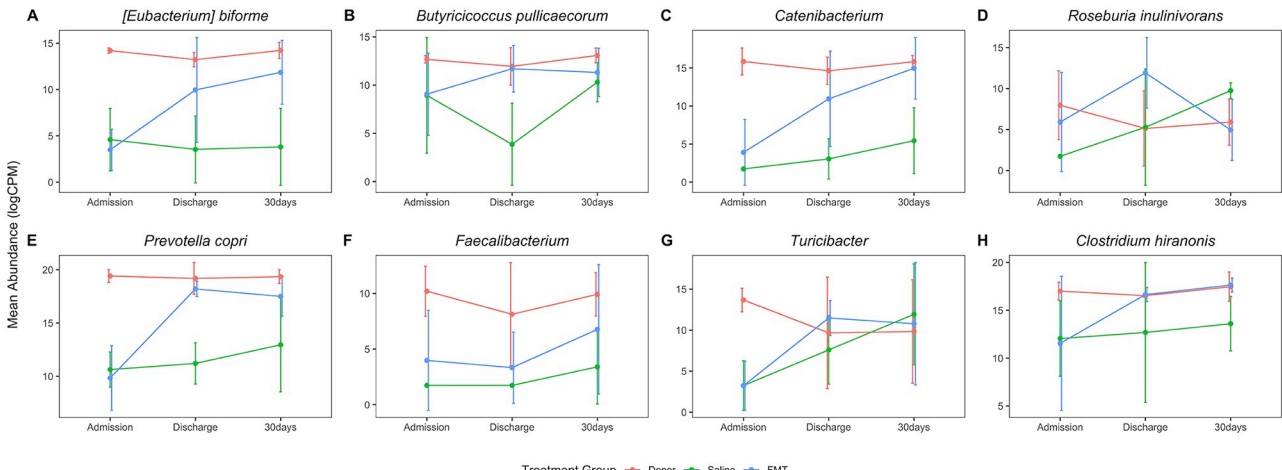

**Fig 7. Changes over time in specific taxa following FMT.** The abundance of each OTU was log-transformed and normalized to counts per million (y-axis) and plotted against the sample collection time (x-axis). Dots and error bars are the mean ± SD. (A-H) The abundance of metabolically important taxa, including short-chain fatty acid and secondary bile acid generating bacteria, increased following FMT. CPM: counts per million; FMT: fecal microbial transplantation.

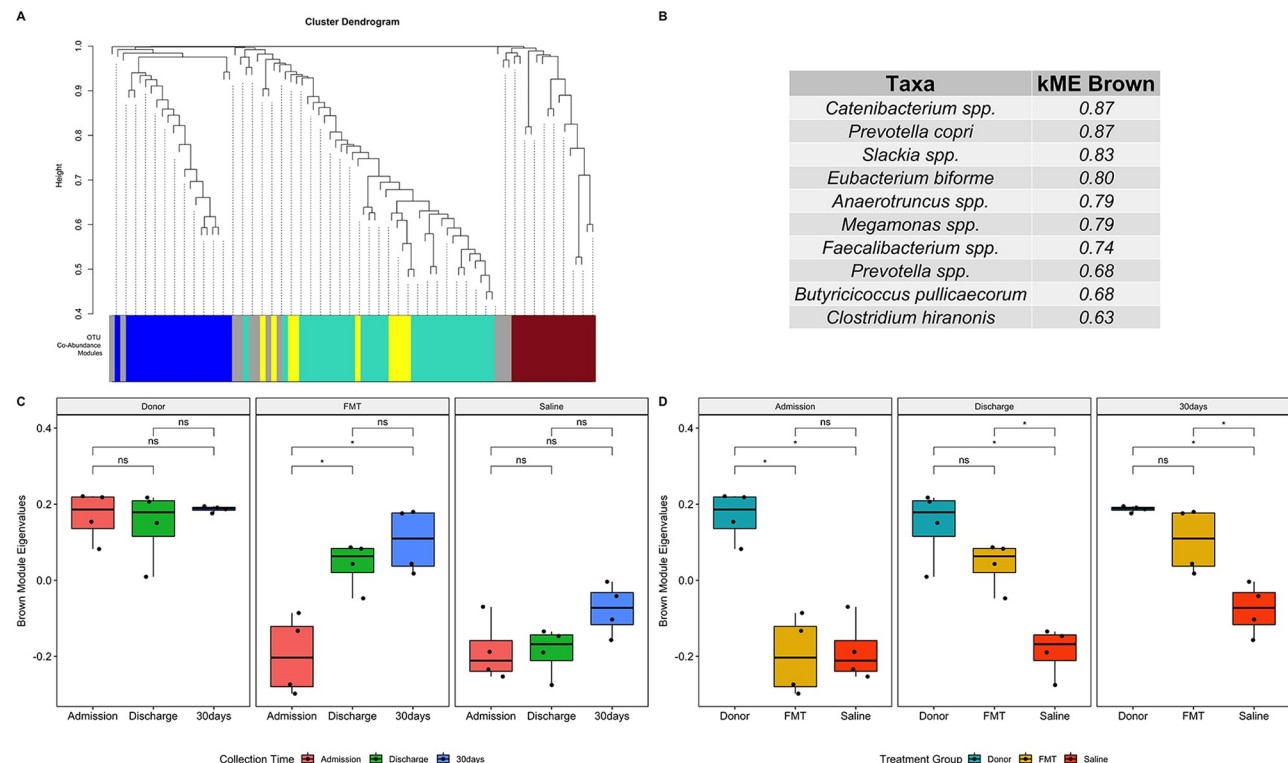

**Fig 8. Weighted co-abundance analysis identified a subnetwork of bacteria significantly associated with FMT.** A weighted co-abundance network was generated to identify subnetworks (modules) of taxa associated with FMT. Module eigen-OTUs (a mathematical summary of module abundance derived from principal component analysis) were used to detect differences in module abundance among treatment groups and collection times. (A) A cluster dendrogram of OTUs with correlated patterns of abundance. The OTUs were grouped into 6 modules and represented by an arbitrarily unique color below the dendrogram. OTUs that were not grouped into a co-abundance module are represented by the gray color. (B) The "Brown" module was found to contain the same taxa that were differentially abundant in post-FMT samples. The importance of each OTU within the "Brown" module was quantified by the module eigen-OTU connectivity (kME), also known as module membership. As the kME of a given OTU approaches 1, it more closely characterizes the abundance pattern of the whole module. OTUs with kME > 0.5 are shown. (C) The "Brown" module was increased at discharge and 30 days in FMT recipients but was unchanged in sham-treated controls. (D) "Brown" module abundance was decreased in the FMT and sham-treated control groups at admission. The "Brown" module increased in abundance in FMT recipients at discharge and 30 days, compared with sham-treated controls. There were no statistically significant differences in "Brown" module abundance between donors and FMT-recipients at discharge or 30 days. Statistical comparisons among treatment groups and collection times are indicated by annotated brackets where "ns" represents P $\geq$ 0.05 and "*" represents P 0.01 < P < 0.05. FMT: fecal microbial transplantation.

significantly increased the abundance of the SCFA-producing bacteria (*Eubacterium biforme*, *Catenibacterium*, and *Prevotella copri*) 30 days post-discharge in the FMT group compared with sham-treated controls (Fig 6A, Table 5) and the higher abundance of these SCFA-producing bacteria in FMT recipient dogs did not differ from the donors (Fig 6A). Our results are similar to those of a longitudinal study that followed the microbiome of human patients with RCI who received FMT [15]. In that study, FMT increased the abundance SCFA-producing bacteria (*Lachnospiraceae incertae sedis*, other *Lachnospiraceae*, *Dorea*, *Peptostreptococcaceae incertae sedis*, *Ruminococcaceae*, and *Subdoligranulum*) [15]. Hence, we cautiously suggest that a single FMT procedure in dogs with AHDS might have the potential to restore specific bacterial communities that are known to be important for intestinal health up to 30 days post-FMT (Figs 6–8, Tables 4 and 5) [46, 53, 54].

Some of the bacteria that were persistently decreased in dogs with AHDS (*Turicibacter*, *Prevotella copri*, *Faecalibacterium* and *Catenibacterium*), transiently decreased (*Roseburia inulinivorans* and *Butyricicoccus pullicaecorum*), or persistently increased (*Turicibacter*, *Prevotella*

*copri* and *Catenibacterium*) in abundance following FMT deserve further attention. These SCFA-producing bacteria predominantly produce butyrate (while *Prevotella copri* produces succinate [55]) [56–62] which is known to be important for intestinal health [46]. Decreased abundance of bacteria such as *Faecalibacterium* and *Turicibacter* has been associated with enteric disease [63, 64] whereas an increase in the abundance of *Catenibacterium* has been associated with decreased lifetime cardiovascular disease risk [65]. However, changes in the content of gut SCFA cannot be predicted by changes in abundances of SCFA-producing bacteria by itself, and necessitates quantitative and qualitative analyses of fecal SCFA, and for the substrates required for SCFA production. Interestingly, changes in the abundance of some of these bacteria have also been linked to diseases affecting tissues outside of the gastrointestinal tract. For example, recent studies indicated a strong link between the decreased production of the bacterial-derived SCFA, butyrate, and disease states such as pre-diabetes and diabetes mellitus in people [66–68]. These findings are mirrored in cats with diabetes mellitus, whose microbiomes have also been found to harbor lower abundances of butyrate-producing bacteria [69].

This pilot study has a few limitations. Firstly, while our a priori analysis indicated that the study was adequately powered to detect a 2-day difference in discharge between FMT and control group, in reality the difference in early discharge following FMT (if present at all) is likely to be of a few hours and thus requiring a much larger sample size. While this is a limitation, we argue that our results are valuable because there are no other comparable studies of how FMT affects the fecal microbiome of dogs with dysbiosis and AHDS. Furthermore, our results could have implications for other diseases (both intestinal and extra-intestinal) associated with dysbiosis. Secondly, the authors acknowledge that not controlling for the diets of the dogs included in the study (both donors and AHDS dogs) is a limitation of the study, as the effect of FMT in a given patient is likely a complex interaction between the host genetic makeup and immune system, host microbiome, donor microbiome, and diet–both preceding and following FMT. However, we argue that if FMT were to be widely used in a clinical setting, controlling the diet would not be possible for practical reasons. We also argue that while the diets differed between dogs, the AHDS dogs clustered together before FMT and significantly diverged in accordance to FMT at the time of discharge (Fig 4). Since everything other than FMT was held constant, we argue that the significant change in community composition was mostly contributed by FMT. Thirdly, because there was only one donor-recipient pair from New Zealand, we could not analyze the possible bias that it might have introduced to the results. While we acknowledge that as a limitation, the New Zealand FMT recipient-donor pair behaved similarly to the other 3 FMT recipient-donor pairs from South Africa (Fig 2B). Also, our results show that donor dogs' microbiota clustered closely together indicating that there was only low variability in their microbiome diversity. Fourthly, as we did not measure fecal concentrations of SCFAs, thus we were unable to determine whether the increase in SCFA-producing bacteria was coupled with an increase in SCFA production. Whilst we acknowledge this as a limitation, there is a strong basis of literature that supports the supposition that the changes in abundance of SCFA-producing bacteria found in this study is related to changes in intestinal and systemic health. Lastly, our study design did not include a control group of AHDS dogs that did not undergo any treatment to ascertain what level of changes in microbial diversity was related to the colon cleansing procedure [70].

## Conclusions

In this small pilot study FMT had no clinical benefit, however, a single FMT procedure resulted in a significant change in the fecal microbiome of FMT recipient dogs up to 1-month

post-FMT. This change was characterized by increased abundance of SCFA-producing bacteria that, according to the literature, are beneficial for the overall intestinal health. Nonetheless, the clinical significance of the bacteriome changes noted in the present study remains to be determined.

## Supporting information

**S1 Fig. Study design and timeline.** Following inclusion of dogs with AHDS to the study and collection of their feces, the dogs were randomized to undergo a colonoscopy procedure during which they received either FMT or sham treatment with saline. The following day, the dogs with AHDS had a colonoscopy procedure. A portion from each of the donors' feces used for FMT was saved. Feces were re-collected from the donors, FMT-recipient dogs with AHDS and sham-treated dogs with AHDS at the time of discharge and 30 days after discharge. AHDS: acute hemorrhagic diarrhea syndrome; FMT: fecal microbial transplantation. (TIF)

**S2 Fig. Heatmap of the differentially abundant taxa associated with FMT clustered by dog level.** AHDS: acute hemorrhagic diarrhea syndrome; FMT: fecal microbial transplantation. (PDF)

**S1 Table. Criteria for clinical assessment of acute hemorrhagic diarrhea syndrome clinical score.** (DOCX)

**S2 Table. Forward and reverse primers used for the amplification of the V3-V4 hypervariable region of the 16S ribosomal RNA gene.** (DOCX)

**S3 Table. Complete results of the Bayesian linear model of differential abundance.** (CSV)

## Acknowledgments

The authors want to thank Dr. Chris Fields (director) and Dr. Jenny Drnevich (functional genomics bioinformatics specialist) from the Center for High Performance Computing in Biology (HPCBio) at the Roy J. Carver Biotechnology Center, University of Illinois at Urbana-Champaign for their oversight and assistance with bioinformatic computation. We thank the reviewers for their helpful suggestions.

## Author Contributions

**Conceptualization:** Arnon Gal, Patrick J. Biggs, Kristene R. Gedye, Anne C. Midwinter, Richard K. Burchell.

**Data curation:** Arnon Gal, Patrick C. Barko, Patrick J. Biggs, Richard K. Burchell, Paolo Pazzi.

**Formal analysis:** Arnon Gal, Patrick C. Barko, Patrick J. Biggs, Kristene R. Gedye, Anne C. Midwinter, Paolo Pazzi.

**Funding acquisition:** Arnon Gal, Patrick J. Biggs, Kristene R. Gedye, Anne C. Midwinter, Richard K. Burchell, Paolo Pazzi.

**Investigation:** Arnon Gal, Richard K. Burchell, Paolo Pazzi.

**Methodology:** Arnon Gal, Patrick J. Biggs, Kristene R. Gedye, Anne C. Midwinter, Paolo
Pazzi.

**Project administration:** Arnon Gal, Patrick C. Barko, Paolo Pazzi.

**Resources:** Arnon Gal.

**Writing – original draft:** Arnon Gal, Patrick C. Barko, Patrick J. Biggs, Kristene R. Gedye,
Anne C. Midwinter, David A. Williams, Richard K. Burchell, Paolo Pazzi.

**Writing – review & editing:** Arnon Gal, Patrick C. Barko, Patrick J. Biggs, Kristene R. Gedye,
Anne C. Midwinter, David A. Williams, Richard K. Burchell, Paolo Pazzi.

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
