## [Decision Letter · Decision Letter 0]

31 Dec 2020

PONE-D-20-34770

One dog’s waste is another dog’s wealth:  a pilot study of fecal microbiota transplantation in dogs with acute hemorrhagic diarrhea syndrome

PLOS ONE

Dear Dr. Gal,

Thank you for submitting your manuscript to PLOS ONE. After careful consideration, we feel that it has merit but does not fully meet PLOS ONE’s publication criteria as it currently stands. Therefore, we invite you to submit a revised version of the manuscript that addresses the points raised during the review process.

The two reviewers agree that the study design and analyses are solid and the manuscript valuable despite the small samples number. The authors should consider all these comments and revise accordingly

We look forward to receiving your revised manuscript.

Kind regards,

Franck Carbonero, PhD

Academic Editor

PLOS ONE

Journal Requirements:

2. Please confirm that the authors obtained consent from owners to have their dogs participate in the research reported in this paper.

'This study was funded by the Massey University’s College of Science Research Fund (MURF). The funders had no role in study design, data collection and analysis, decision to publish, or preparation of the manuscript.'

We note that one or more of the authors are employed by a commercial company: Small Animal Medicine Specialist Services, Townsville.

Additional Editor Comments:

Reviewers are in agreement that these data, while small in scope, is an important addition to the field.

I am copying Reviewer 2's comments here:

This study describes the clinical and faecal bacteriome characteristics of a series of dogs that presented with acute haemorrhagic diarrhea, and were treated with a faecal microbial transplant or saline placebo. The understanding of bacterial disturbances associated with, and causing disease (dysbiosis) in dogs is foetal at best, and barely embryonic in regards to the response to FMT. This study, though very small, describes changes to the bacteriome in clinical cases, and represents an important contribution to a very empty landscape. The study was well designed (with one significant criticism), is methodologically strong, well presented, and whilst the analysis bordered on excessive given then small number of dogs, it was conducted to a high standard. It is very unlikely that we will develop a general understanding of the contribution of the bacteriome to health and disease for some time to come, and it will require the assimilation of a large number of studies that include a wide variety of dogs, diets, and clinical diseases, and as such, this present study will undoubtably become part of that canon.

Major criticisms

I have two major criticisms that I would appreciate the authors address. One is conceptual and relates more to the discussion and interpretation, and the other is a methodological limitation. Firstly, the authors refer to, and base their analysis on the concept that AHDS is a syndrome with features that are common to most, if not all cases. The manuscript starts, not with a description of AHDS or a defense of it as a unitary disease, but with a description of recurrent C.difficile enterocolitis in humans. It may warrant comparison with the disease in humans at some point in the discussion, but in this reviewer’s opinion they are not sufficiently similar as to justify comparison in the introduction. That would be a small point were it not for the fact that it sets the tone for manuscript and the interpretation of the data. The dogs included in this study, and indeed in others that have preceded it, do not have recurrent disease, nor do they have overgrowth of a clostridial species, let alone difficile. In fact, on face value, this study more strongly suggests that AHDS is not caused by a dysbiosis than that it is. I would want the authors to consider these dogs as sharing some common clinical features, but certainly not necessarily a common aetiology, and thus one would not expect there to be a common cure. Whilst I do not claim that a microbial disturbance is not causal in AHDS, I claim that a) there are many potential causes of the syndrome, b) the abnormalities described in this paper may well be secondary and not causal, and c) the findings of this paper provide evidence against a microbial hypothesis, are discordant with previous studies, and certainly do not support any similarity with C.diff infections in humans.

Secondly, the intestinal microflora is the product of bacteria that are introduced into it, and the substrate that is provided, and modified by host and community factors. The authors have only considered the bacteria introduced through the FMT, and have not reported, or apparently considered the effects of substrate. Numerous studies have shown the profound effect of diet on the bacteriome, and it is very likely that any sustained change effected by FMT in a given patient will be a complex interaction between host, disease, host microflora, donor microflora, and diet – both preceding and following transplantation. I am sure that other readers will be surprised that no effort was made to either control, or even describe the post-treatment diet, and it is a major oversight not to consider its effects.

Specific criticisms

Line 50: It is no clear what the relevance of the comparison with RCI is, especially in light of the author’s own data which disprove any similarity.

Line 62: I disagree with general statements regarding the “dysbiosis of AHDS”. Two studies have shown some differences, though the two cited studies unsurprisingly differed. In keeping with the lack of conformity, none of the taxa that were most different in the study by Suchodolski et al, were altered in the author’s study. Thus, their own data prove that there is no a generalizable feature of the bacteriome in this syndrome. In fact, it remains to be seen if the clinical presentation of these dogs has a common aetiology, and if it does, it does not appear to be due to consistent changes in the bacteriome.

Line 73: Why would the authors not have hypothesized that FMT would have a clinical benefit? I find it hard to believe that this study was conducted with the intention simply to look at the faecal bacteria, and it has the sense of post hoc hypothesis creation, since the treatment had no clinical benefit. I think to avoid suspicion of such post hoccery, the authors need to either explain why a clinical benefit was of no interest, or include that hypothesis, which after all, the authors do go on to test anyway. If no clinical benefit was expected, what is the justification of the study other? And given that most dogs with idiopathic AHDS recover rapidly anyway with supportive care, what sort of clinical benefit were the authors trying to detect? What case numbers would be necessary to be able to detect a small difference in time to improvement?

Line 82 and 85: Please change these written numbers “Seven and 1”. It sounds like something from the old testament!

Table 1. It is interesting that one of the donor dog’s was consuming Hill’s metabolic. Was the dog obese? That is a very high fibre diet compared with the other diets, and is known to modify the bacteriome compared with a standard diet. Why was the French Bulldog being fed z/d? Did it have a prior history of diarrhea? It is disappointing that there was no dietary history for most of the recipients. The authors cannot say if differences at the time of admission are not due to dietary differences, and be unrelated to disease. Whilst I do not suspect that, it cannot be excluded, and should be discussed.

Line 93: Were faeces graded? How severe was the diarrhea?

Line 94: Please better describe the disease definition. How was “haemorrhagic diarrhoea” diagnosed? What constituted “haemoconcentration”?

What other differential diagnoses were considered and excluded? Addisons? Toxicity? Pancreatitis? How was endoparasitism excluded?

Line 104: The authors are probably aware of the rapid shifts in viable bacteria, especially anaerobic species, following defaecation, a process that is greatly accelerated with delays in handling, and inadvertent aeration of the samples. The reader requires more detail in the process of faecal handling. What quantity of faeces was collected? Were they placed immediately into RNAlater? Also, there is stratification of species from mucosa to lumen. When swabs or loops are used to collect samples, there is a bias towards mucosally-adherent species, when compared with the faecal bacteriome. Thus, when collected directly from the rectum, the nature of the specimen is influenced by the rectal contents. How was that considered/safeguarded against?

Line 105-106: Please clarify if samples from dogs with AHDS were collected in a manner that differed from donor dogs or not.

Line 112: AHDS is, as the authors note, a syndromic diagnosis, and not one that can be confirmed on histopathology. In addition, a variety of histopathologies have been described depending on the cause and severity.

Line 210: It seems a little disingenuous to suggest that there was any alternative to using more than a single donor given the two study sites!

Line 128: How did the authors define “healthy”. It is a nebulous concept, but at least the reader should know what boundaries the authors put on it.

Line 133: Same comment as for line 104. More detail please. The authors should especially comment on the effect of the processing on the viability of oxygen-sensitive species.

Line 144: What pre/post operative fasting/feeding protocol was used for the dogs?

What was the post-treatment diet? Surely that was standardized? At the very least, that needs to be reported, as any differences will be heavily influenced by colonic substrate, and perhaps all differences were due to dietary differences.

Line 152: placed “immediately” into a cryovial? Was this after the processing (mixing, sieving, etc)? If it was not the same, then please discuss the potential for a) differences between bacteriome and viable bacteria, and b) differences between the bacteriome as described from this sample, and differences that can occur ex-vivo between defaecation and transplantation.

Line 158: Presumably the pathologist was blinded to group allocation. Please specify.

Line 273: I’m not clear on what statistical test was used to examine the significance of the AHDS scores. It seems that there was a comparison between groups at each time point, and between timepoints within each group, but what about an overall model that accounts for baseline value? Friedman’s test?

Line 277: I would have wanted more clinical detail about these dogs. The AHDS scores are presented, but what was the range of clinical signs? What were their absolute Hct values, and did they differ between groups?

Line 280-281: Faecal grades?

Line 287: Was some sort of histopathological grade or score applied? If there was variation in severity and types of histological changes (e.g. regeneration, nature of infiltrate), then it would be interesting to see if they were associated with any particular bacteriome. I realise that this is a very small study, but it might be the basis of further hypothesis formulation.

The fact that some dogs had an eosinophilic infiltrate is demonstrative of the heterogeneity of this syndrome, and probably the variation in aetiology. This speaks against it being a homogenous dysbiosis that would be expected to be treated successfully with an essentially random FMT microflora.

Line 375: Given that this is not a defined disease, with a standard dysbiosis, and that clearly there was heterogeneity of disease in these dogs, focusing on the common differences ignores the possibility that there are pathogenic species / combinations in individuals. There is little the authors can do to analyse the data to avoid that, but perhaps there should be more discussion of that possibility.

Table 3. The authors really do need to note that they are trying to force a holistic understanding on a heterogeneous collection of starting bacteriomes, different aetiologies, and hence different disturbances. The authors cite the study by Suchodolski et al, but they fail to mention that not one of the “significantly different taxa” that defined AHDS in that study (e.g. Sutterella sp, C.perfringens, Ruminococcae, Blautia, etc), were significantly different in this present study, and vice versa! That is the key point to emphasise: the authors appear to be attempting to create a bacteriological definition of the disease and hence suggest a common treatment, yet that is, from the authors’ own data, demonstrably not the case.

Line 391-397: I don’t think it’s appropriate to make claims about the benefits of changes in a few genera of bacteria in relation to their metabolic products. The impact on the host is dependent on the entire microflora, and the substrate provided through the diet. There are overall quantitative and qualitative effects that completely supersede these relative changes. One bacteria may utilize one substrate, which may then be utilized to produce another substrate by another bacteria. The authors may be able to conjecture about SCFA production if they looked at the entire population in each dog, considering both the species and their absolute abundances, and then considered the population outputs given a common dietary substrate, but we shouldn’t make weak speculations based on a few relative changes.

Line 435: “referral hospital population”.

Line 442: It does not follow that simply because the histological scores were low, that the dogs were already recovering. And it was always going to be the case that it would be hard to demonstrate a clinical benefit, that authors surely knew that at the onset.

Line 446-447: Please avoid suggesting this is a single common disease. You are conflating what may be many different aetiologies.

Line 449-450: It is not appropriate to discuss this as if they are the same, as these dogs are different.

Line 453: AHDS is not “a defined thing”, that can have an effect on the microflora. And surely the authors are arguing in reverse – that AHDS is caused by a dysbiosis?

Line 454: Here the authors again are paltering with the concept that AHDS is a single entity. They cite the papers by Suchodulski and Markel, supposedly in support of a common dysbiosis, but fail to point out the more startling differences.

Line 463: Are these simply epiphenomena, given the absence of any clinical difference or benefit?

Line 474-475: You cannot infer that as a conclusion: any effect of the FMT is entirely dependant on the microflora present in the donor. The benefits of an FMT may in the future be proven to be universal, as long as faecal quality is normal in the donor, but at the moment that seems very unlikely, and the examples of dysbiosis in otherwise apparently healthy dogs with normal faeces are legion. Thus, we will need, in the future, to define an appropriate donor microflora on the basis of the microflora, and not simply on the basis of the clinical appearance of normality.

Lines 478-490: Again, the SCFA faecal concentrations cannot be predicted from looking at differences in a few genera. If the authors want to discuss this, and they are right to consider it, then they need to consider each individual dog’s entire microflora quantitatively and qualitatively, and then consider that in light of the diet being fed, and only then speculate as to what the faecal SFCA, let alone colonic luminal concentrations might be.

Conclusions

I would have expected the authors to start with the conclusion that, in this small study, FMT had no clinical benefit. It should also include the statement that the significance of the bacteriome changes noted are of no known significance, given that this is not analogous to recurrent C.diff colitis in humans.

Figures – I hope that text will be easily readable in all figures. I did not find Figure 7 useful.

Reviewers' comments:

Reviewer's Responses to Questions

**Comments to the Author**

1. Is the manuscript technically sound, and do the data support the conclusions?

Reviewer #1: Yes

2. Has the statistical analysis been performed appropriately and rigorously? 

Reviewer #1: Yes

3. Have the authors made all data underlying the findings in their manuscript fully available?

Reviewer #1: Yes

4. Is the manuscript presented in an intelligible fashion and written in standard English?

Reviewer #1: Yes

5. Review Comments to the Author

Reviewer #1: In this manuscript entitled “One dog’s waste is another dog’s wealth: a pilot study of fecal microbiota transplantation in dogs with acute hemorrhagic diarrhea syndrome” the authors aimed to determine the effect of FMT on composition of fecal microbiota in dogs with AHDS. This study consists of a small number of AHDS dogs treated with FMT (n=4) and a control group (n=4). The manuscript is well written and the statistical analysis is robust. In its current state, this paper has limited clinical utility since no clinical improvement was observed with FMT treatment, however it provides valuable information about the microbial composition of dogs suffering from AHDS.

Major revisions:

1. There is a known association of AHDS and C. perfringens infection (https://www.ncbi.nlm.nih.gov/pmc/articles/PMC6505910/). This manuscript obtained histopathology but did not perform testing for C. perfringens and/or the presence of netF. To strength this paper, it is recommended that this information be added for the AHDS dogs. This helps to ensure the classification as AHDS.

2. In the 16S data, there is no mention of C. perfringens, it is surprising that none of the 8 dogs with AHDS would not have had this organism present in the analysis? Was this OTU identified and just not mentioned? It is recommended that this be added to the manuscript as well.

3. More information is needed for the FMT donor dogs. Was fecal infectious disease testing performed on donor dogs (such as the IDEXX Real PCR diarrhea panel)? Infectious disease testing in dogs should be standard of care and this is unclear in this manuscript.

4. Preparation for colonoscopy can have an effect a profound effect on the GIT (https://www.nature.com/articles/s41598-019-40182-9). Since the aim of the paper was to characterize the fecal microbiota of AHDS during disease and recovery it would have been helpful to include a group of AHDS that did not receive colonoscopy but underwent the same in hospital treatments.

5. In-hospital treatment algorithms are not mentioned. This study would be strengthened by controlling what other treatments dogs received in the hospital. Additional information regarding treatments doing hospital stay and discharge is needed. Dietary history from hospital stay and discharge is also needed. Did the diet remain consistent throughout the study period for all AHDS dogs and donors?

6. Added discussion needed about using 4 different donor dogs for this study and the variability that is seen overtime in donor dogs.

7. The authors compare AHDS to recurrent C. difficile infection (rCDI) in people. I do not feel this is a good comparison, since AHDS in most dogs is self-limiting and rCDI is not. It is recommended that this be de-emphasized in both the introduction and discussion as these diseases are not as similar as the authors described.

8. One dog with AHDS was on a diet that clinically is used in dogs with diet responsive chronic enteropathy (z/d), can the authors please provide additional information about this dog’s GI disease/history?

9. Was fecal scoring (Purina and/or Waltham) completed? This should be included for both the donors and AHDS dogs at admission, discharge, and 30 days post.

10. Figure 3: Changing these to traditional box and whisker plots (with individual data points within the box) would be helpful.

11. Figure 6B. This heatmap is confusing. It may be helpful to perform unsupervised clustering on dogs instead of the OTUs, this would allow the reader to visualize that AHDS dogs were most similar to their donor.

12. In the discussion, it is recommended to include more information on the beneficial effects of restoring the microbiota (such as potential to prevent chronic enteropathy, etc.) since a clinical benefit was not observed in this small pilot.

13. In the conclusions, be specific that no immediate clinical benefit (such as faster resolution of diarrhea, faster recovery, shorter hospitalization, etc.) was observed.

Minor Revisions:

1. Line 50: Clostridium difficile is the incorrect and outdated name, this needs to be corrected (https://pubmed.ncbi.nlm.nih.gov/27370902/)

2. Line 399: Correct “OUT” to “OTU”

3. Line 403: Be specific about which potential pathobionts were observed

4. Line 447: Should be specific and list C. perfringens here.

5. Lines 483-485: Be specific about which species these changes were documented in.

6. S1 Table: Appetite and defecation are misspelled

6. PLOS authors have the option to publish the peer review history of their article (what does this mean?). If published, this will include your full peer review and any attached files.

Reviewer #1: No

---

## [Author Response · Author response to Decision Letter 0]

28 Feb 2021

Our response to the reviewers was attached as a separate file.

---

## [Decision Letter · Decision Letter 1]

6 Apr 2021

One dog’s waste is another dog’s wealth: a pilot study of fecal microbiota transplantation in dogs with acute hemorrhagic diarrhea syndrome

PONE-D-20-34770R1

Dear Dr. Gal,

We’re pleased to inform you that your manuscript has been judged scientifically suitable for publication and will be formally accepted for publication once it meets all outstanding technical requirements.

Kind regards,

Franck Carbonero, PhD

Academic Editor

PLOS ONE

Additional Editor Comments (optional):

Reviewer 2 was not able to submit their comments online, so here they are:

I’d like to thank the authors for their thoughtful arguments and considered emendations. The introduction is greatly improved, and addresses the uncertainty, the bacterial heterogeneity, and better describes the rationale for the FMT in these dogs. I note that, as is often the case, when changes are made in response to a review, the inserted portions lack the grammatical rigor of the original. I would recommend a final proof reading prior to resubmission.

Line 90: I still dispute that “Seven and 1” is a sensible turn of phrase, and it would be much better to write “Eight dogs……., of which 7 were from X and 1 was from Y”.

Line 102: since Hct is a physical parameter, it is not lab specific, whereas lab ranges are. Please provide the actual cut off value used by the authors. It is implied in the results, but should be given here as it is part of the diagnostic criteria.

In response to my criticism of the speculation that SCFA concentrations are increased (Lines 478-490 in the original manuscript), the authors have made what seems to me as an appropriate adjustment. As an aside, the authors suggest that clinically significant changes in faecal metabolite concentrations can be predicted by the bacteriome without consideration of other factors, notably the diet, which I still question. In response, the authors cite the study by Zierer et al. I argue that there is a large difference between studying variation in the faecal metabolome using an untargeted approach, and making a claim about specific metabolites. In addition, PLS-DA accounts for variation, but by design does not account for the actual concentrations. Lastly, in the study by Zierer et al, SFCAs were not amongst those that were most strongly associated, so I don’t find the citation supportive of the claim, and my criticism remains. This requires no answer in the manuscript from the authors, but is a thought worth pondering.

The authors suggest a significant discordance in view, though I’m not sure my point of view is significantly different. In fact, unless the authors now don’t agree with their own revisions, I would suggest we are in complete agreement on the major points!

The different criticisms from both reviewers can be simply resolved: Reviewer #1 makes the claim that this is a study of dogs with AHDS, which may or may not have shared aetiologies, but all dogs with AHDS probably have different aetiologies. Reviewer #2 makes the point that C.perf net-F toxin is one cause of AHDS, which I do not dispute. Whether there is value in determining if any of these particular dogs are toxin-positive is not clear to me, especially in light of not having found large numbers of C.perfringens amongst the OTUs.

Reviewers' comments:

Reviewer's Responses to Questions

**Comments to the Author**

1. If the authors have adequately addressed your comments raised in a previous round of review and you feel that this manuscript is now acceptable for publication, you may indicate that here to bypass the “Comments to the Author” section, enter your conflict of interest statement in the “Confidential to Editor” section, and submit your "Accept" recommendation.

Reviewer #1: All comments have been addressed

2. Is the manuscript technically sound, and do the data support the conclusions?

Reviewer #1: Yes

3. Has the statistical analysis been performed appropriately and rigorously? 

Reviewer #1: Yes

4. Have the authors made all data underlying the findings in their manuscript fully available?

Reviewer #1: Yes

5. Is the manuscript presented in an intelligible fashion and written in standard English?

Reviewer #1: Yes

6. Review Comments to the Author

Reviewer #1: Thank for addressing the comments from the review. The changes to the introduction are clear and have greatly improved this manuscript!

Minor edits below:

Need to ensure that all 16S is publicly available via NCBI SRA.

Corrections that are required:

• Line 24: C. perfringens was not renamed only C. diff. This needs to be changed to Clostridium perfringens.

7. PLOS authors have the option to publish the peer review history of their article (what does this mean?). If published, this will include your full peer review and any attached files.

Reviewer #1: No

---

## [Editor Report · Acceptance letter]

8 Apr 2021

PONE-D-20-34770R1 

One dog’s waste is another dog’s wealth: a pilot study of fecal microbiota transplantation in dogs with acute hemorrhagic diarrhea syndrome 

Dear Dr. Gal:

I'm pleased to inform you that your manuscript has been deemed suitable for publication in PLOS ONE. Congratulations! Your manuscript is now with our production department. 

Kind regards, 

on behalf of

Dr. Franck Carbonero 

Academic Editor

PLOS ONE